# Trojan Horse nanotheranostics with dual transformability and multifunctionality for highly effective cancer treatment

Xiangdong Xue [1], Yee Huang[1,2], Ruonan Bo[1], Bei Jia[1], Hao Wu[1], Ye Yuan[1], Zhongling Wang[1,3], Zhao Ma[1], Di Jing[1,4], Xiaobao Xu[1], Weimin Yu[1,5], Tzu-yin Lin[6] & Yuanpei Li [1]

Nanotheranostics with integrated diagnostic and therapeutic functions show exciting potentials towards precision nanomedicine. However, targeted delivery of nanotheranostics is hindered by several biological barriers. Here, we report the development of a dual size/charge- transformable, Trojan-Horse nanoparticle (pPhD NP) for delivery of ultra-small, full active pharmaceutical ingredients (API) nanotheranostics with integrated dual-modal imaging and trimodal therapeutic functions. pPhD NPs exhibit ideal size and charge for drug transportation. In tumour microenvironment, pPhD NPs responsively transform to full API nanotheranostics with ultra-small size and higher surface charge, which dramatically facilitate the tumour penetration and cell internalisation. pPhD NPs enable visualisation of biodistribution by near-infrared fluorescence imaging, tumour accumulation and therapeutic effect by magnetic resonance imaging. Moreover, the synergistic photothermal-, photodynamic- and chemo-therapies achieve a 100% complete cure rate on both subcutaneous and orthotopic oral cancer models. This nanoplatform with powerful delivery efficiency and versatile theranostic functions shows enormous potentials to improve cancer treatment.

[1] Department of Biochemistry and Molecular Medicine, UC Davis Comprehensive Cancer Center, University of California Davis, Sacramento, CA 95817, USA. [2] Institute of Animal Husbandry and Veterinary Science, Zhejiang Academy of Agricultural Sciences, Hangzhou, Zhejiang 310021, China. [3] Department of Radiology, Shanghai General Hospital, Shanghai Jiao Tong University School of Medicine, Shanghai 200080, China. [4] Department of Oncology, Xiangya Hospital, Central South University, Changsha, Hunan 410008, China. [5] Renmin Hospital of Wuhan University, Wuhan, Hubei 430060, China. [6] Division of Hematology/Oncology, Department of Internal Medicine, University of California Davis, Sacramento, CA 95817, USA. These authors contributed equally: Xiangdong Xue and Yee Huang. Correspondence and requests for materials should be addressed to Y.L. (email: lypli@ucdavis.edu)

Nanotheranostics are engineered to simultaneously achieve non-invasive diagnosis and effective therapeutics in a single nanoformulation[1–3], which not only provides promising potentials to real time visualise the ADMET (absorption, distributions, metabolism, elimination and toxicity)[4] of the nanomedicine, but also realises personalised, synergistic and accurate therapeutic functions in a visualised manner. Although promising, various biological barriers severely affect the delivery efficiency of nanotheranostics into tumour tissue and cells. The challenges mainly include: (i) undesirable circulation time in blood; (ii) insufficient tumour accumulations due to inefficient tumour vascular extravasation; (iii) limited tumour tissue penetrations due to the high interstitial fluid pressure and dense extracellular matrix in the tumour microenvironment (TME); (iv) ineffective cell internalisation and (v) inefficient intracellular drug release. It is typically difficult to circumvent these obstacles simultaneously. For instance, PEGylated nanoparticles could lessen the opsonization[5,6], and enhance their blood circulation time[7–9], therefore provide more chances for nanoparticles to take advantages of enhanced permeability and retention (EPR) effect in solid tumour[10]. However, the surface charge of the PEGylated nanoparticles is close to neutral, which may mitigate the cell internalisation, because of the inefficient interaction with the negatively charged glycocalyx[11] on the cell surface. The positively charged nanoparticles are readily for cell uptake, but vulnerable to the opsonization, and thereby, being rapidly cleared from blood. The nanoparticles with smaller size have more chance to overcome the interstitial transport hindrance and diffuse deeply into tumour tissue[12]. However, they may be less desirable for tumour vascular extravasation and favourable pharmacokinetics. In addition, if the size of these nanoparticles is less than 5 nm, they are likely to be excreted through kidney[3]. The nanoparticles with a larger size may be able to circumvent the renal clearance and have more advantages in tumour vascular extravasation, but their tumour penetration is typically unsatisfactory.

Recently, several transformable nano-systems have been developed to address the above-mentioned challenges[13–16]. For instance, Chen's group developed a gene delivery system to balance the surface charge issues for improvement of both tumour accumulation and cell uptake[17]. Their group also developed charge/size dual-transformable nanoparticles for overcoming a variety of delivery barriers[18]. Wang's group reported clustered nanoparticles to realise size transformability, and significantly improved the tumour penetration and anti-tumour efficacy[19]. However, these strategies were mainly with mono-therapeutic effect only, and lacked diagnosis capacities. Tumours are usually heterogeneous, which often makes the treatments complicated, and their responses to monotherapy approach are insufficient in many clinical cases[20]. Construction of nanotheranostics with synergistically combined multi-therapeutic functions and more sophisticated design to circumvent the multiple biological barriers for greatly improved delivery efficiency are highly desirable for complete tumour elimination. Furthermore, the integrated imaging functions of nanotheranostics offer unique capabilities in non-invasively monitoring the real-time therapeutic delivery and assessment of treatment outcomes.

Here, we report a dual size/charge- transformable, Trojan-Horse nanoparticle (pPhD NP) for targeted delivery of ultra-small, multifunctional, full active pharmaceutical ingredient (API) nanotheranostics with integrated dual-modal imaging and trimodal therapeutic functions that enable to achieve multimodal imaging-guided tumour eradication. As illustrated in Fig. 1, the construction of pPhD NPs starts from a self-assembly of amphiphilic molecules (PhD monomer), which are synthesised by the conjugation of two API components, a hydrophobic photosensitizer (pheophorbide a, Pa) and a hydrophilic anti-neoplastic drug (doxorubicin, DOX),

through hydrazone bonds that are cleavable at acidic intracellular pH (pHi). The PhD monomers first assemble into micelle-like, ultra-small, full API nanotheranostics, and then further assemble into relatively large nanovehicles (upPhD NPs). Dual-aldehyde terminated polyethylene glycol$_{2000}$ (PEG-2CHO) is introduced and reacted with the amine groups on the surface of upPhD NPs and concurrently cross-link the nanoparticles through the formation of Schiff base bonds that are cleavable at acidic extracellular pH (pHe) of tumour, resulting in the final pPhD NPs. The pPhD NPs are tailored to overcome various drug delivery barriers and integrate versatile theranostic functions simultaneously in a single formulation: (i) The PEGylation could significantly reduce the positive surface charge to improve the blood circulation time, and concurrently form intraparticle cross-linkages to stabilise the nanoparticles; (ii) The PEG surface could be responsively detached at pHe of TME to release PhD-based nanotheranostics with ultrasmall size and highly positive surface charge, and thereby, dramatically improve the tumour tissue penetration and overall cell internalisation; (iii) The hydrazone bond between Pa and DOX could be cleaved at pHi inside the lysosomes of tumour cells to further accelerate the drug release; (iv) The ultra-small nanotheranostics with nearly 100% API contents (photosensitizer/chelator and chemo-drug) could synergistically combine phototherapy and chemotherapy to improve the efficacy; (v) The intrinsic optical- and magnetic-resonance-imaging capabilities of Pa in the nanoparticles could be used to visualise the real-time in vivo delivery and therapeutic efficacy non-invasively.

## Results

**Construction and characterisations of PhD NPs.** The PhD monomers were synthesised by conjugating pheophorbide a (Pa) and doxorubicin (DOX) through a pHi cleavable hydrazone bond (Supplementary Figure 1). The results from mass spectrometry and nuclear magnetic resonance (NMR) studies indicated that the PhD monomers were successfully synthesised (Supplementary Figure 2–5). The synthesis and characterisation of dual-aldehyde terminated PEG$_{2000}$ (PEG-2CHO), and its reaction with amine groups on DOX was illustrated in Supplementary Figure 6–9. The PhD monomers were first assembled into PhD NPs, then PEGylated and cross-linked by PEG-2CHO through the formation of Schiff base, resulting in PEGylated PhD NPs (pPhD NPs, as shown in Fig. 1). The particle morphology was observed by TEM (Fig. 2a). The pPhD NPs were shown as spherical nanostructures, in which hived hundreds of small dark dots. As illustrated in Fig. 1, this Trojan-Horse liked nano-architecture was a secondary self-assembly of small PhD micelles through multi-micelle aggregation[21–24]. The size distribution of pPhD NPs was measured by dynamic light scattering (DLS). As shown in Fig. 2b, pPhD NPs were around 79 nm, and the polydispersity index (PDI) was 0.2. In pPhD NPs, the content of DOX was ~24.9% (mass ratio) while that of photosensitizer (Phy, Phy is Pa with a hydrazide pendant) was ~28.4% (mass ratio) (Supplementary Figure 10). The critical aggregation concentrations (CAC) of pPhD NPs were calculated to be 3 μM (Supplementary Figure 11). The nanoparticle stability was monitored by incubating the pPhD NPs in the presence of serum. As shown in Supplementary Figure 12, pPhD NPs retained their original size distribution for 2 weeks. The UV–vis spectra (Fig. 2c) of PhD monomers showed elevated absorbance of DOX around 488 nm and Phy peak around 412 and 670 nm, indicating the PhD monomer contained both Phy and DOX. The fluorescence spectra (Fig. 2d) showed that the emission of DOX was at ~590 nm, and that of Phy was at ~680 nm. While being conjugated together, the fluorescence of DOX at 590 nm decreased, and that of Phy at 680 nm increased, indicating a fluorescence resonance energy transfer may occur in

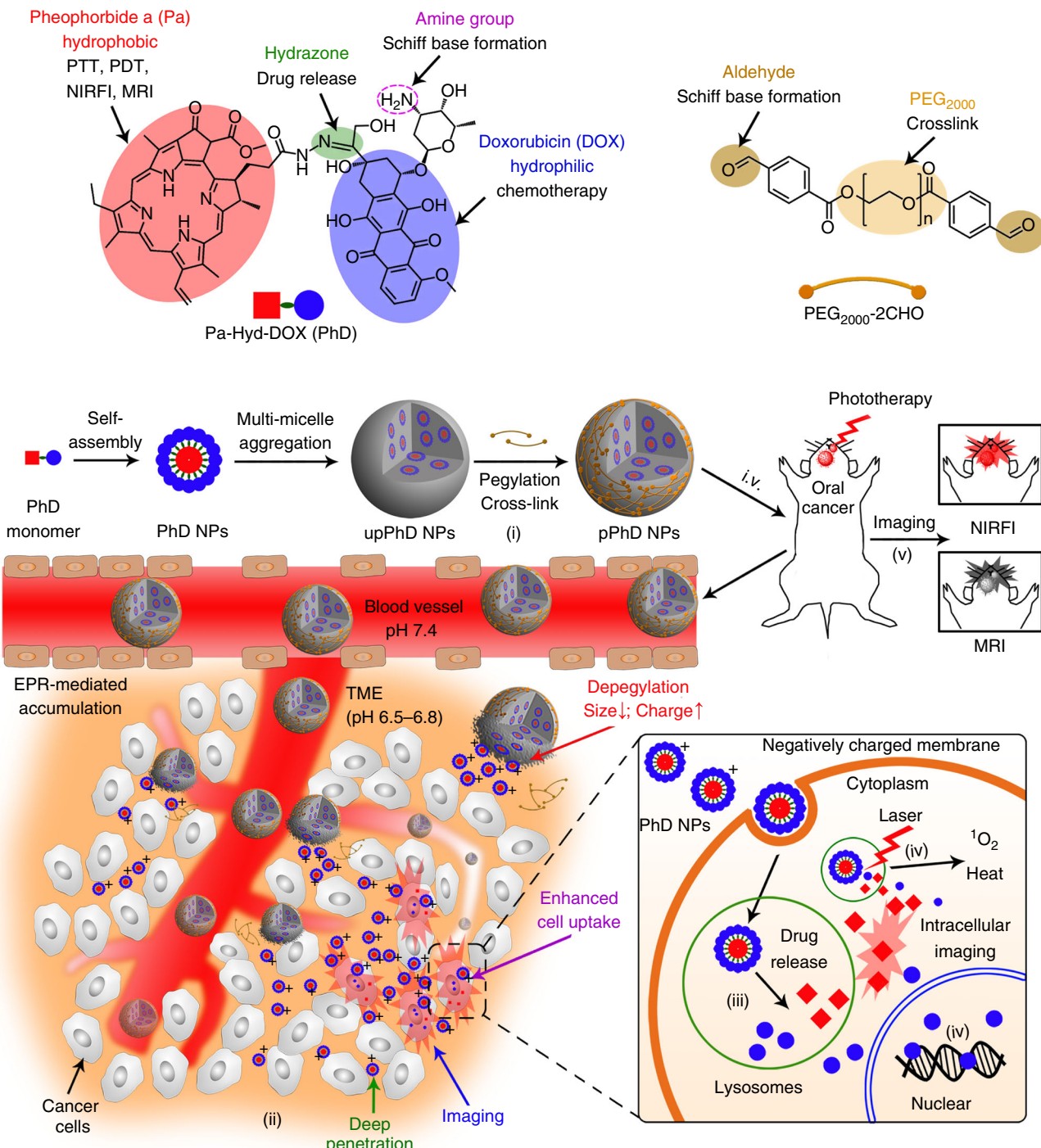

**Fig. 1** Schematic illustration of the Trojan-Horse nanoparticles. Image illustration of the functionalities of the building blocks and construction of the Trojan-Horse nanoparticles (pPhD NPs), and their capabilities to overcome a series of biological barriers through TME responsiveness, de-PEGylation/cross-linkage, transformability, superior penetrations, enhanced cell internalisation, intracellular drug release for multimodal imaging and trimodality therapies. Abbreviations: PDT photodynamic therapy, PTT photothermal therapy, NIRFI near-infrared fluorescence imaging, MRI magnetic resonance imaging, TME tumour microenvironment, Red square denotes Phy, and blue circle denotes DOX, PhD pheophorbide a-hydrazone-doxorubicin, upPhD NPs unPEGylated PhD NPs, pPhD NPs PEGylated PhD NPs, $^1O_2$ reactive oxygen species

PhD monomers. In nanoformulation (pPhD NPs), aggregation caused quenched (ACQ) phenomenon[25,26] dominated and quenched both fluorescences of Phy and DOX (Fig. 2e).

**Near-infrared imaging, photothermal and photodynamic effect of pPhD NPs**. Since porphyrin derivatives are intrinsically suitable for near infrared imaging (NIRFI)[27–29], the NIRFI capacity of the PhD monomers and its nanoformulation (pPhD NPs) was evaluated in an animal imaging system. The PhD monomer exhibited excellent fluorescence signal (Fig. 2f), indicating it was appropriate for NIRFI. The pPhD NPs showed very low fluorescence due to the occurrence of ACQ, which was consistent with the results from the fluorescence spectra (Fig. 2e).

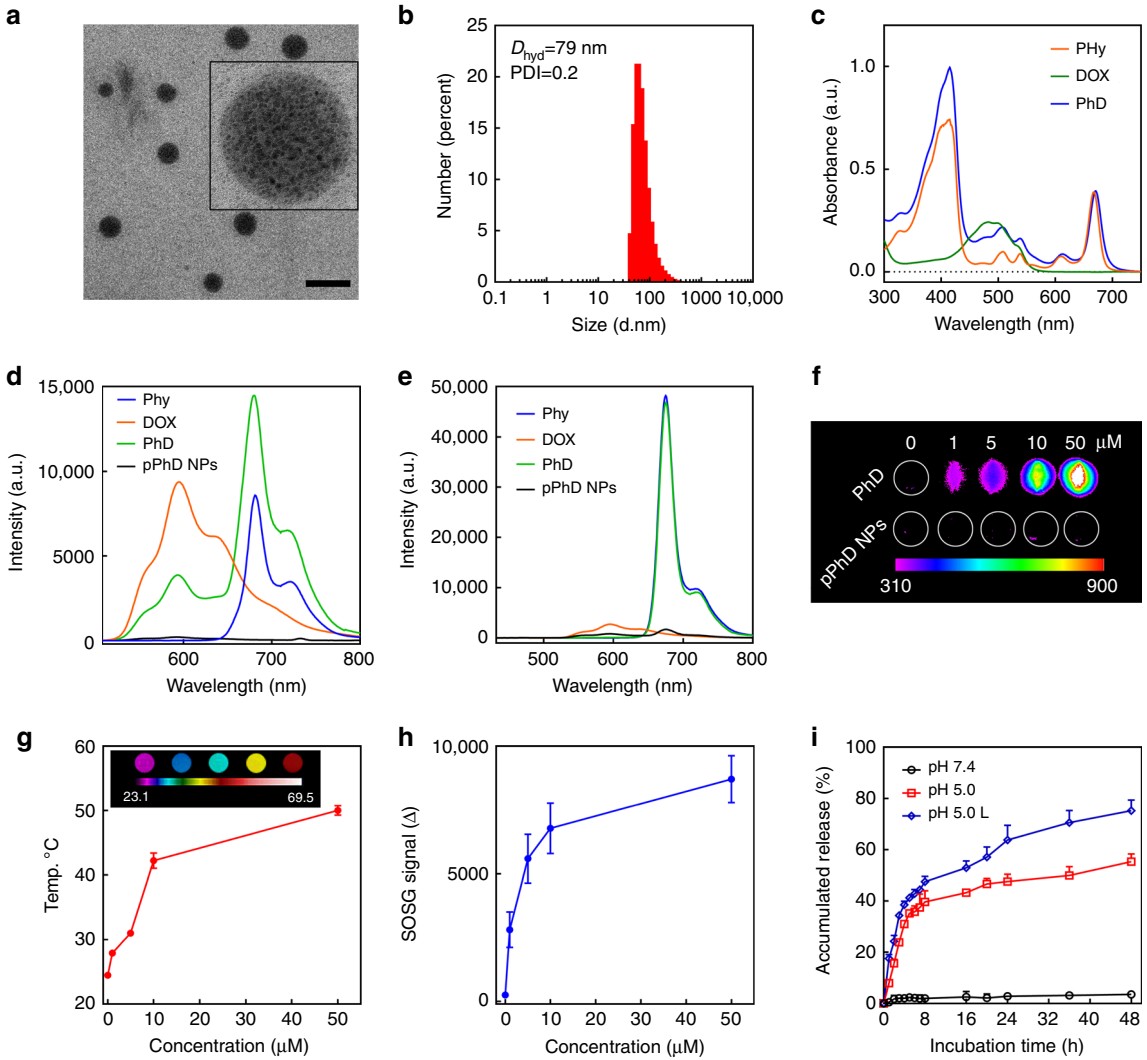

**Fig. 2** Characterisation of pPhD NPs. **a** TEM micrograph of pPhD NPs (50 μM). The scale bar is 100 nm. **b** Size distributions of pPhD NPs (50 μM) measured by dynamic light scattering (DLS). **c** UV–vis absorbance of 50 μM Phy, DOX and PhD monomer. Fluorescent spectra of 5 μM Phy, DOX, PhD and pPhD NPs, **d** excited at 488 nm (optimal excitation of DOX) and **e** excited at 412 nm (optimal excitation of Pa). **f** Near-infrared imaging of PhD monomer and its nanoformulation (pPhD NPs) at different concentrations. The unit of the gradient bar is given as arbitrary unit (a.u.) to present the fluorescence intensity is relatively higher or lower. **g** Photothermal effect and **h** photodynamic effect (ROS production) of 50 μM pPhD NPs measured by a thermal imaging camera and singlet oxygen sensor green (SOSG) as an ROS indicator, respectively ($n = 3$). The inset image in **g** was captured by a thermal imaging camera, the unit of gradient bar is °C. **i** Drug releasing patterns ($n = 3$) of pPhD NPs (100 μM) at pH 7.4 and pH 5.0 with and without laser irradiation. All laser power was set to 0.4 W cm$^{-2}$ and the irradiation time was 3 min. All error bars are presented as standard deviation

Fig. 2g showed that the temperature of pPhD NPs increased to around 50 °C upon laser irritation, demonstrating their excellent photothermal property. Furthermore, the pPhD NPs could produce considerable reactive oxygen species (ROS) in a concentration-dependent manner (Fig. 2h).

**pH-stimulus drug release of pPhD NPs**. The hydrazone bond could be cleaved at pHi inside tumour cells[30–32]. Therefore, the pPhD NPs were designed to release the drug under the stimulation of acidic pH and/or laser. Firstly, the cleavage of the hydrazone bond was proved by incubating PhD monomers in pHi (pH 5.0) and subjected to liquid chromatography-mass spectrometry (LC-MS). Hydrazone bond is well known to be traceless reversible chemical bond[31,33], and cleavage at hydrazone bond will yield two original chemicals that were employed to construct the PhD monomer. Hence, three peaks were expected to appear in the total ion chromatography (TIC) spectrum,

including the peaks of Phy (~607 Da), DOX (~544 Da), and un-cleaved PhD monomer (~1132 Da). As shown in Supplementary Figure 13, the TIC spectrum exhibited three main peaks (upper panel) and each peak corresponded to a mass spectrum (lower panel). The doxorubicin was first eluted out (#1, 4.70 min), and the corresponding mass spectrum showed peak at 544.1 Da; the second peak was the PhD monomers (#2, 6.56 min), and was correlated to 1132.5 Da in mass spectrum; and the last TIC peak (#3, 8.54 min) was related to 607.2 Da, indicative of the Phy. The LC-MS results supported that the PhD molecules can be cleaved by pHi from hydrazone bond. Then, the accumulated drug releasing pattern of pPhD NPs was shown in Fig. 2i. The nanoparticles were stable in physiological pH with minimal drug release. The release could be significantly accelerated in acidic pH (5.0, mimicking the lysosomes pH) that closed to pHi. While triggered with both laser and acidic pH, the nanoparticles could release the drug even faster and the accumulated drug release rate reached nearly 80% within 48 h. The drug-releasing pattern

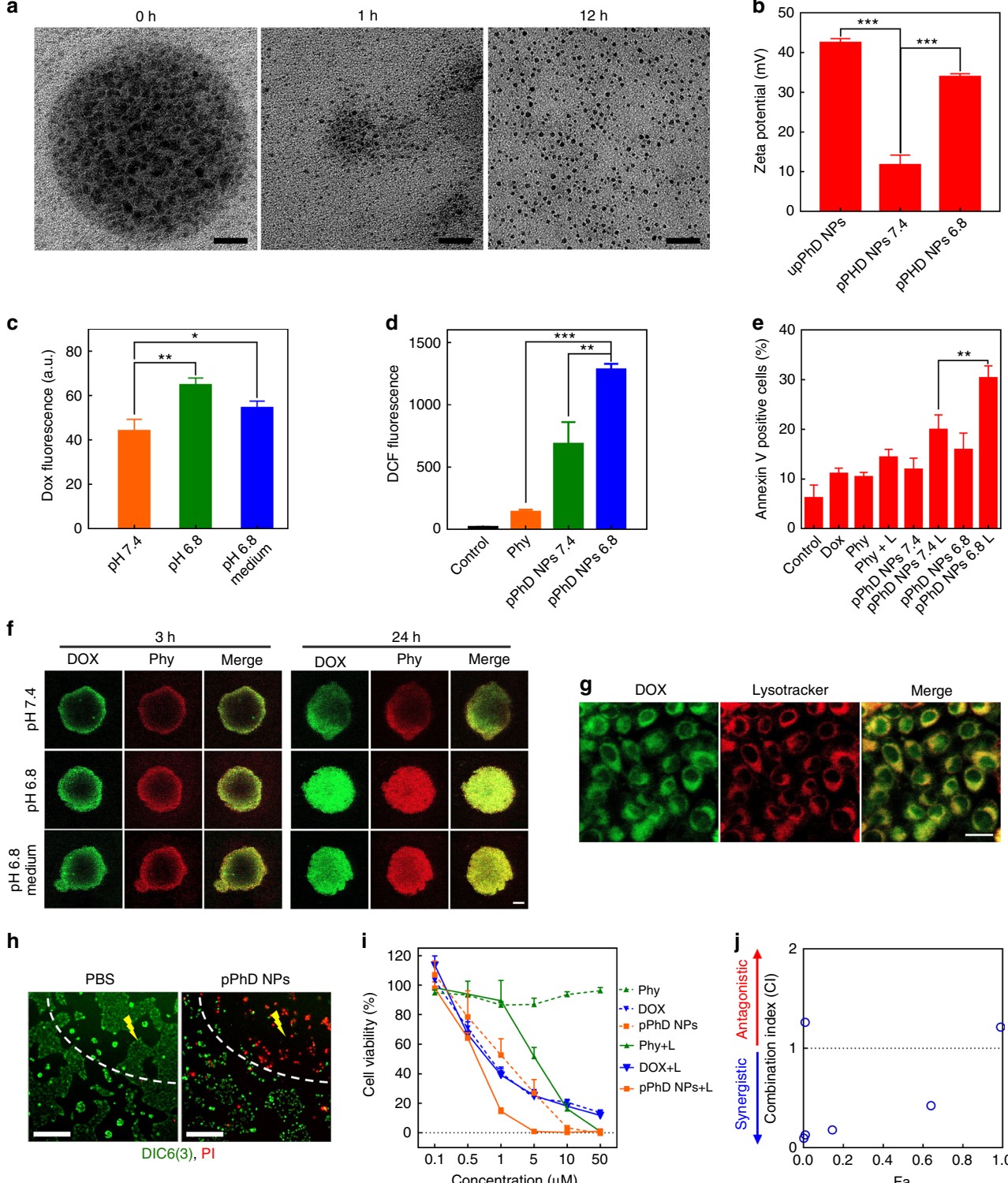

supported that the pPhD NPs could stay stable in physiological conditions, but effectively release the drug under specific stimuli (pH and/or laser).

**Size/charge dual-transformability of pPhD NPs.** We hypothesised that the ultra-small nanoparticles that constrained in pPhD NPs would be released after the peeling of the PEG surface in the TME, as the Schiff base used for PEGylation/cross-linkage was ultrasensitive to pHe[17,34]. To prove this hypothesis, pPhD NPs

were incubated at pH 6.8 for different time, and their Trojan-Horse liked behaviours were directly observed by TEM (Fig. 3a). At the very beginning, pPhD NPs were stable, and able to accommodate hundreds of ultra-small nanoparticles in the Trojan-Horse liked nanostructure. While the pPhD NPs could still be observed at 1 h, most ultra-small nanoparticles were released. At 12 h, all ultra-small nanoparticles (~4 nm) were released (Fig. 3a). The TEM micrographs demonstrated that the pPhD NPs were stable enough to retain the ultra-small nanoparticles under normal physiological condition but could

**Fig. 3** In vitro evaluation of pPhD NPs. **a** TEM micrographs illustrated the Trojan-Horse liked behaviours of pPhD NPs (50 μM) at pH 6.8. The scale bar of 0 h micrograph is 10 nm; 1 h and 12 h are 30 nm. **b** Surface charge changes ($n = 3$) of the pPhD NPs (50 μM) before/after PEGylation and de-PEGylation. De-PEGylation was realised by incubated pPhD NPs at pH 6.8. **c** Cell uptake ($n = 3$) of the pPhD NPs (10 μM) before and after being transformed at pH 6.8; the medium was adjusted to pH 6.8 to assess the dynamic transformation-enhanced cellular uptake. **d** ROS production of the nanoparticles in OSC-3 cancer cells ($n = 3$). **e** Apoptosis of OSC-3 cells ($n = 3$). In ROS and apoptosis analysis, the concentrations of materials, including photosensitizer, free DOX and nanoparticles, were all set as 10 μM. **f** Cell penetration evaluated in tumour cell spheroids. the medium was adjusted to pH 6.8 to assess the dynamic transformation-enhanced superior penetration. The scale bar is 100 μm. **g** Lysosomes co-localisation illustrated that drug was released in lysosomes. The scale bar is 20 μm. **h** Laser-directed phototherapeutic effect on cells, the yellow lighting symbol indicates laser-treated area. The live cells were indicated by DIC6(3), the dead cells stained with PI. **i** Cell viabilities of OSC-3 cells by treated with different concentrations of Phy, DOX and pPhD NPs, with or without light irradiation ($n = 3$). **j** Combination index (CI) of chemotherapy and phototherapy with pPhD NPs towards OSC-3 cells. $*p < 0.05$; $**p < 0.01$; $***p < 0.001$. All error bars are presented as standard deviation

effectively release ultra-small nanoparticles in response to pHe in TME. The changes in surface charges further confirmed the PEGylation and de-PEGylation (Fig. 3b). Before PEGylation, the nanoparticles (upPhD NPs, un-PEGylated PhD NPs) exhibited strongly positive charge (43 mV). While after PEGylation, the surface charge decreased to 12 mV. When the pPhD NPs were treated at pH 6.8, the charge rebounded to 35 mV. The results of TEM and surface charge studies supported that pPhD NPs were dual-transformable, both size and surface charge could be transformed to desirable values that may be beneficial to superior tumour penetration (ultra-small size) and enhanced cell uptake (strong positive charge).

**Transformability-enhanced cellular uptake, ROS production and apoptosis in vitro.** Furthermore, we investigated the benefits of the dual-transformability of pPhD NPs in oral squamous cell carcinoma 3 (OSC-3) cells. pPhD NPs (pH 7.4) and post-transformed pPhD NPs (pre-treated in pH 6.8) were incubated with OSC-3 cells to evaluate the transformability-enhanced cell uptake. pPhD NPs incubated with cells in slightly acidic cell culture medium (pH 6.8 medium) was also employed to explore the benefits of dynamic transformability. As shown in Fig. 3c. the post-transformed pPhD NPs and the dynamic transformed groups both showed significantly enhanced cellular uptake than the non-transformed nanoparticles. The cellular uptake results indicated that the elevated surface charge could significantly enhance the cellular uptake and the slightly acidic pHe would be able to trigger this process as well. We then evaluated the ROS production in OSC-3 cells and found that the post-transformed nanoparticles (pPhD NPs at pH 6.8) produced significantly higher amount of ROS in comparison to free photosensitizer (Phy) and the nanoparticles at pH 7.4 (Fig. 3d and Supplementary Figure 14). The cell apoptosis assays showed consistent results. Post-transformed pPhD NPs exhibited more significant apoptosis than their counterpart at pH 7.4 and other control groups (Fig. 3e and Supplementary Figure 15).

**Tumour penetrations and lysosome-colocalization of the pPhD NPs.** The pPhD NPs could transform to nanoparticles with ultra-smaller size that may penetrate deeper in tumour tissue than the particles with larger size. To prove it experimentally, pPhD NPs and post-transformed pPhD NPs were incubated with OSC-3 cell spheroids, respectively, and observed under confocal microscopy. pPhD NPs incubated with cell spheroids in slightly acidic medium (pH 6.8 medium) was also employed to explore the dynamic transformability-enhanced penetration (Fig. 3f). In pPhD NPs treated tumour cell spheroids, the fluorescence of DOX and Phy were both distributed at the periphery at the first 3 h, then diffused further at a prolonged incubation time (24 h). Upon transforming into ultra-small nanoparticles (pPhD NPs at pH 6.8), the fluorescence signal spread much further than that at

neutral pH at the first 3 h, then diffused throughout the whole tumour spheroid after 24 h incubation. The acidic medium treatment also dynamically led to superior penetration of pPhD NPs than that in neutral medium. This result indicated that the pHe would be able to stimulate the transformability and the resulting ultra-small nanoparticles could penetrate much deeper into the tumour spheroids than the bigger nanoparticles. After the nanoparticles have been ingested into the tumour cells, the pPhD NPs were expected to release the drug (DOX) upon the cleavage of hydrazone bonds by pHi inside the lysosomes. We incubated pPhD NPs with OSC-3 cells and co-localised the fluorescence of DOX (green) with lysosomes (red). As shown in Fig. 3g, DOX showed large co-localisation areas with lysosomes, indicating that our nanoparticles could release DOX in lysosomes, in which the pHi enabled cleavage of the hydrazone bond.

**In vitro controllable phototherapy of pPhD NPs.** We then irradiated a discrete area of OSC-3 cells pre-incubated with pPhD NPs, and observed the laser-treated and non-treated cells (Fig. 3h). Most of OSC-3 cells treated with pPhD NPs & laser were dead as indicated by PI staining, while the cells incubated with pPhD NPs without laser treatment, exhibited much less cell death. As a control group, the PBS treated cells showed no obvious cell death, in both regions exposed or not exposed to laser. These results indicated that the phototherapy with pPhD NPs was controllable, only impacted the region where the laser was directed.

**Synergistic effect of phototherapy and chemotherapy.** The synergistic effect of the chemotherapy and phototherapy of pPhD NPs was evaluated. OSC-3 cells were incubated with different concentrations of free photosensitizer (Phy), free chemotherapeutic drug (DOX) and pPhD NPs, respectively, then treated with or without laser (Fig. 3i). In the non-laser-treated group, free Phy exhibited no obvious cytotoxicity while free DOX and pPhD NPs showed notable anti-tumour efficacy. In the laser-treated group, Phy exhibited enhanced efficacy comparing to the non-laser-treated counterpart. The cell-killing effect of DOX remained at a similar level. It is worth noting that the pPhD NPs treated group showed the most effective anti-tumour activity against OSC-3 cells among all the groups with or without laser treatment. We then calculated the combination index (CI)[35,36] of the phototherapy and chemotherapy based on Fig. 3i, which demonstrated that these therapeutic modalities showed excellent synergistic effect to kill the cancer cells (Fig. 3j and Supplementary Table 1).

**Pharmacokinetics (PK) profiles of the pPhD NPs.** The PEGylation can shield the highly positive surface charge and form the interparticle cross-linkage which may greatly affect the blood circulation and tumour accumulation of pPhD NPs. Therefore,

the un-PEGylated PhD NPs (upPhD NPs as illustrated in Fig. 1) were introduced as control to elucidate the influence of PEGylation on the PK and biodistribution. The morphology and size distributions of upPhD NPs were first investigated. As shown in Supplementary Figure 16, the upPhD NPs exhibited a similar morphology to pPhD NPs, which showed bigger nanoparticles that constructed with clusters of small-dots, but without PEG corona wrapping around. The hydrodynamic size was around 100 nm and slightly larger than pPhD NPs. Then, the PK profiles of pPhD NPs was determined in jugular vein cauterised rats by comparing with the equivalent dose of upPhD NPs and free DOX. As shown in Supplementary Figure 17 and Supplementary Table 2, two nanoformulations, including pPhD NPs and upPhD NPs, possessed similar T-half ($\alpha$) that was much longer (7.2 times) than free DOX, indicating that the nanoformulation enabled to largely mitigate the blood clearance of the chemotherapeutic drugs. The PEGylated nanoparticles (pPhD NPs) showed the longest second phase circulation half-time (T-half ($\beta$), 1440 min), which provided a long time window for tumour accumulation. The pPhD NPs also exhibited bigger AUC and higher C-max than un-PEGylated ones (upPhD NPs), which supported that the PEGylation improved the blood circulation time and minimised the opsonization effect. For upPhD NPs, the C-max was close to free DOX but much lower than pPhD NPs, indicating that substantial amounts of un-PEGylated nanoparticles suffered from opsonization at the first 2 min after i.v. administration, causing the rapid elimination of the nanoparticles. In contrast, free DOX showed weaker PK profile (including C-max, AUC and T-half) than both nanoformulations.

**Biodistribution in orthotopic oral tumour models by NIRFI.** Oral cancer commonly occurs at sites of the lips, tongue, cheeks, floor of the mouth, hard and soft palate, sinuses, and pharynx, and is readily accessible to light. It represents an excellent clinical situation for the potential applications of pPhD NPs developed in this study. We then investigated whether the PEGylation could greatly improve biodistribution profile in orthotopic oral cancer models established by implantation of OSC-3 cells into the lips of nude mice. The in vivo NIRFI of upPhD NPs and pPhD NPs were conducted on orthotopic oral tumour model. As shown in Supplementary Figure 18 & 19, both nanoparticles preferentially accumulated at tumour site. The ex vivo NIRFI further confirmed that their higher accumulation in tumours than in normal organs (Fig. 4a). The fluorescence signal of pPhD NPs in the centre of the tumour was much stronger than that of upPhD NPs. The quantitative fluorescence comparison (Fig. 4b) showed that pPhD NPs exhibited significantly higher tumour accumulation than its un-PEGylated counterpart (upPhD NPs).

**Time-dependent tumour accumulation of pPhD NPs by MRI.** In contrast to optical imaging, MRI has superior features, like deeper penetration. MRI also offers excellent spatial and anatomic resolution. As pPhD NPs have intrinsic capability to chelate manganese (II) ion ($Mn^{2+}$) (Fig. 4c), we could conveniently utilise MRI to visualise the tumour accumulation of the nanoparticles in real time. The UV–vis and fluorescence spectra in Supplementary Figure 20 supported that the $Mn^{2+}$ was successfully chelated to PhD molecules[37,38]. The T1 MRI contrast of PhD NPs was concentration-dependent (Fig. 4d) and the relaxivity ($r^1$) of $Mn^{2+}$ chelated PhD NPs was calculated to be 2.89 $mM^{-1} S^{-1}$. The dynamic contrast-enhanced MR images in orthotopic oral cancer models were displayed in Fig. 4e. The T1-weighed MR signal at tumour sites showed a time-dependent manner with the MR signal intensity increased after injection of the nanoparticle, reached a peak at 24 h, then gradually decreased.

Interestingly, the MR signals of pPhD NPs were significantly higher than the non-transformable nanoformulation (upPhD NPs) at 4, 8, 24, 48, and 72 h post-injection (Fig. 4e, f). The MR signals of pPhD NPs retained at tumour sites at a considerable level for up to 72 h. The PK, NIRFI and MRI studies demonstrated that the PEGylated cross-linkage greatly contributed to the better in vivo performance of pPhD NPs, such as the longer blood circulation and better tumour accumulation.

**Investigation of the in vivo phototherapeutic effects.** The phototherapeutic effects of pPhD NPs were investigated on the orthotopic oral cancer model. As shown in Fig. 4g, the photosensitizer-harboured groups all exhibited better photothermal effect than PBS control as measured by the temperature rises at the tumour site. Among these groups, pPhD NPs exhibited highest heat generation, and the temperature of the tumours treated with pPhD NPs increased about 24 °C. Fig. 4h, i displayed the photodynamic effects, in which pPhD NPs treated group produced significantly more ROS production than other three groups.

**Phototherapeutic outcomes of pPhD NPs visualised by MRI.** As $Mn^{2+}$ chelated pPhD NPs possessed intrinsic capacity as a MRI contrast agent, the $Mn^{2+}$ chelated pPhD NPs was i.v. injected into orthotopic oral tumour-bearing mice to in situ monitor their phototherapeutic outcomes by MRI (Fig. 4j). The T1 MRI signal dramatically increased at 24 h after the administration of pPhD NPs. We then applied two doses (at 24 and 48 h post-injection) of laser to trigger the phototherapeutic effect, and the orthotopic oral tumour was continuously monitored for changes in size and morphology. In the MR images, the tumour shrunk extensively at 72 h and became smaller and smaller with time elapse. The majority of tumour was ablated at 7 days (168 h) of post-injection. The MRI visualisation showed promising merits for evaluation of the therapeutic effects that cannot be observed by naked eyes, especially for a tumour that cannot be directly reachable.

**In vivo therapeutic effects of pPhD NPs.** We further performed systematic treatment studies in both subcutaneous and orthotopic tumour models to verify the synergistic therapeutics and superior efficacy of pPhD NPs. The OSC-3 cells were implanted to two positions of the flanks or lips of nude mice to establish subcutaneous and orthotopic tumour models, respectively. After tumour formation at 15 days, the mice were randomly assigned into 5 groups ($n = 6$): control (PBS), free drug (DOX), free photosensitizer (Phy), un-PEGylated PhD NPs (upPhD NPs) and PEGylated PhD NPs (pPhD NPs). The experimental design and workflow of the animal studies were shown in Supplementary Figure 21. All tumour-bearing mice were treated once per week for three consecutive weeks by i.v. administration. In subcutaneous models (mice bearing two tumours), the right tumours that treated with photosensitizer-harboured materials were subjected to laser exposure (0.4 W $cm^{-2}$, 3 min), and the left-side tumours were not treated with laser to evaluate the efficacy of chemotherapy alone (Fig. 5a). In the orthotopic models, all tumours treated with photosensitizer-harboured materials were treated with laser (Fig. 5b). The laser treatments were given twice at 24 and 48 h after the i.v. injection. Tumour volumes and body weights were measured throughout the treatments. The changes in tumour volume of the subcutaneous model were shown in Fig. 5c. Since the oral cancer is highly malignant, the PBS and free photosensitizer (without laser) groups didn't exhibit obvious antitumour efficacy. The tumour grew fast and all mice in these two groups were sacrificed (considered dead) due to the oversized

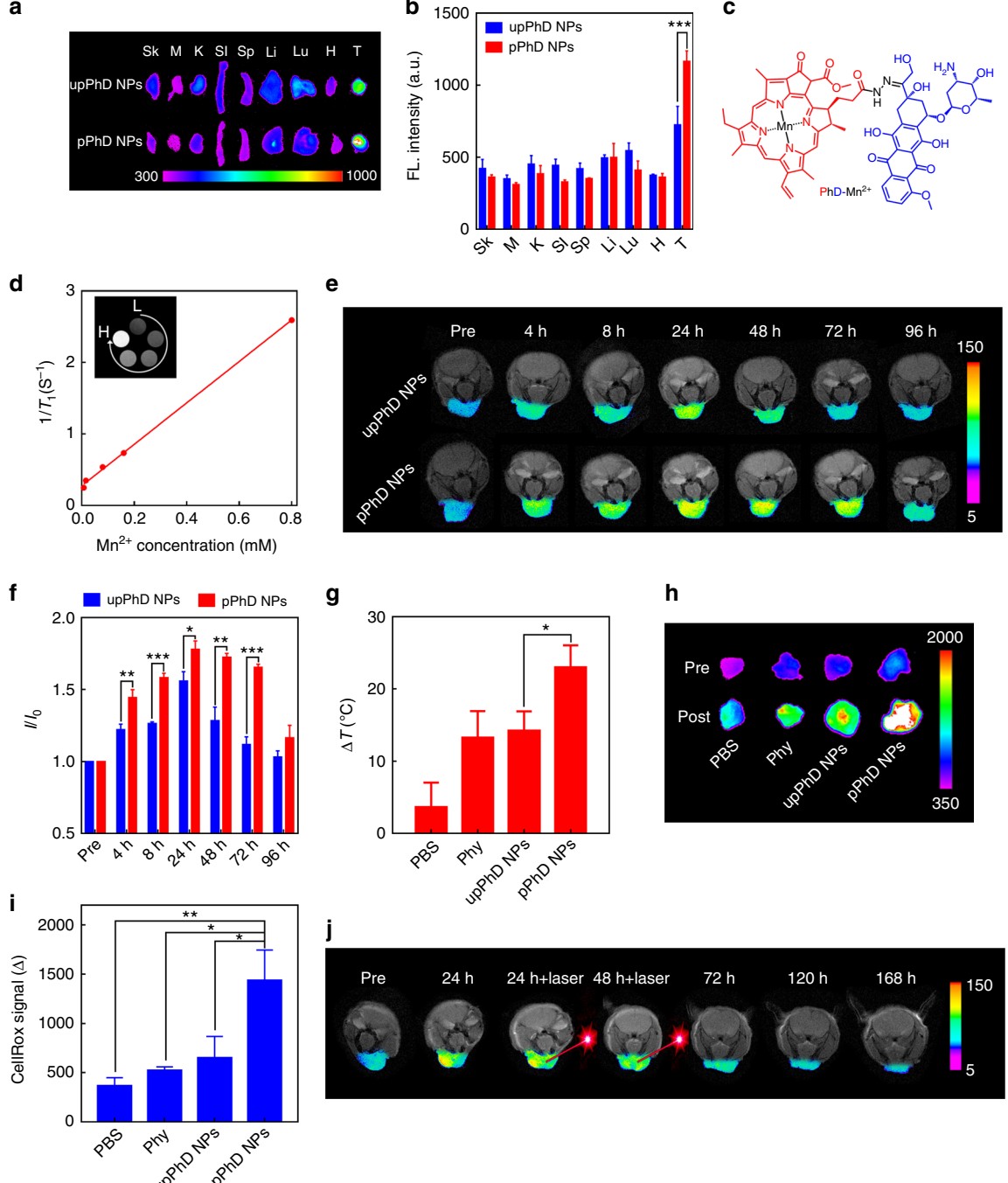

**Fig. 4** In vivo evaluation of pPhD NPs in orthotopic oral cancer models. **a** NIRFI images and **b** quantitative fluorescence with statistical analysis of the ex vivo distributions ($n = 3$) of the nanoparticles. **c** Chemical structures illustrated the chelation of the manganese(II) ions ($Mn^{2+}$) to PhD molecules. **d** Concentration-dependent relaxation of $Mn^{2+}$ chelated pPhD NPs. The $Mn^{2+}$ chelated in pPhD NPs were 0.008, 0.016, 0.08, 0.16 and 0.8 mM, respectively. L denotes the low concentrations started at 0.008 mM, H denotes the high concentrations ended at 0.8 mM. **e** T1-weighted MRI images of time-dependent tumour accumulations ($n = 3$) of the nanoparticles acquired on a 7 T MRI scanner and **f** the quantitative MR signal intensity changes ($I/I_0$) on orthotopic oral cancer model. $I$ is MR signal at a specific timepoint, $I_0$ is the MR signal of the mice at Pre timepoint. Pre denotes the mice before $Mn^{2+}$ chelated pPhD NPs treatment. **g** Photothermal effects of the nanoparticles on orthotopic oral cancer model ($n = 6$). The laser (680 nm) dose was 0.4 W cm$^{-2}$ for 3 min. **h** Fluorescence imaging of ROS productions within tumour tissues and **i** quantitative comparisons with statistical analysis of different treatments on orthotopic oral cancer model ($n = 3$). The laser (680 nm) dose was 0.4 W cm$^{-2}$ for 3 min. The ROS productions were indicated by NIRF ROS probe, CellROX. Pre denotes the NIRFI before CellROX treatment; Post means the NIRFI after CellROX indication. **j** Phototherapeutic effect monitored by MRI. The laser (680 nm) dose was 0.8 W cm$^{-2}$ for 3 min. Pre denotes the MRI of the mice before treatment with $Mn^{2+}$ chelated pPhD NPs. For all animal experiments above, the injection doses of upPhD and pPhD NPs were 10 mg kg$^{-1}$ (calculated based on the concentration of PhD monomer). Phy was 5.3 mg kg$^{-1}$. *$p < 0.05$; **$p < 0.01$; ***$p < 0.001$. The units of the gradient bars are all given as arbitrary unit (a.u.) to present the fluorescence or MRI signal intensities are relatively higher or lower. All error bars are presented as standard deviation

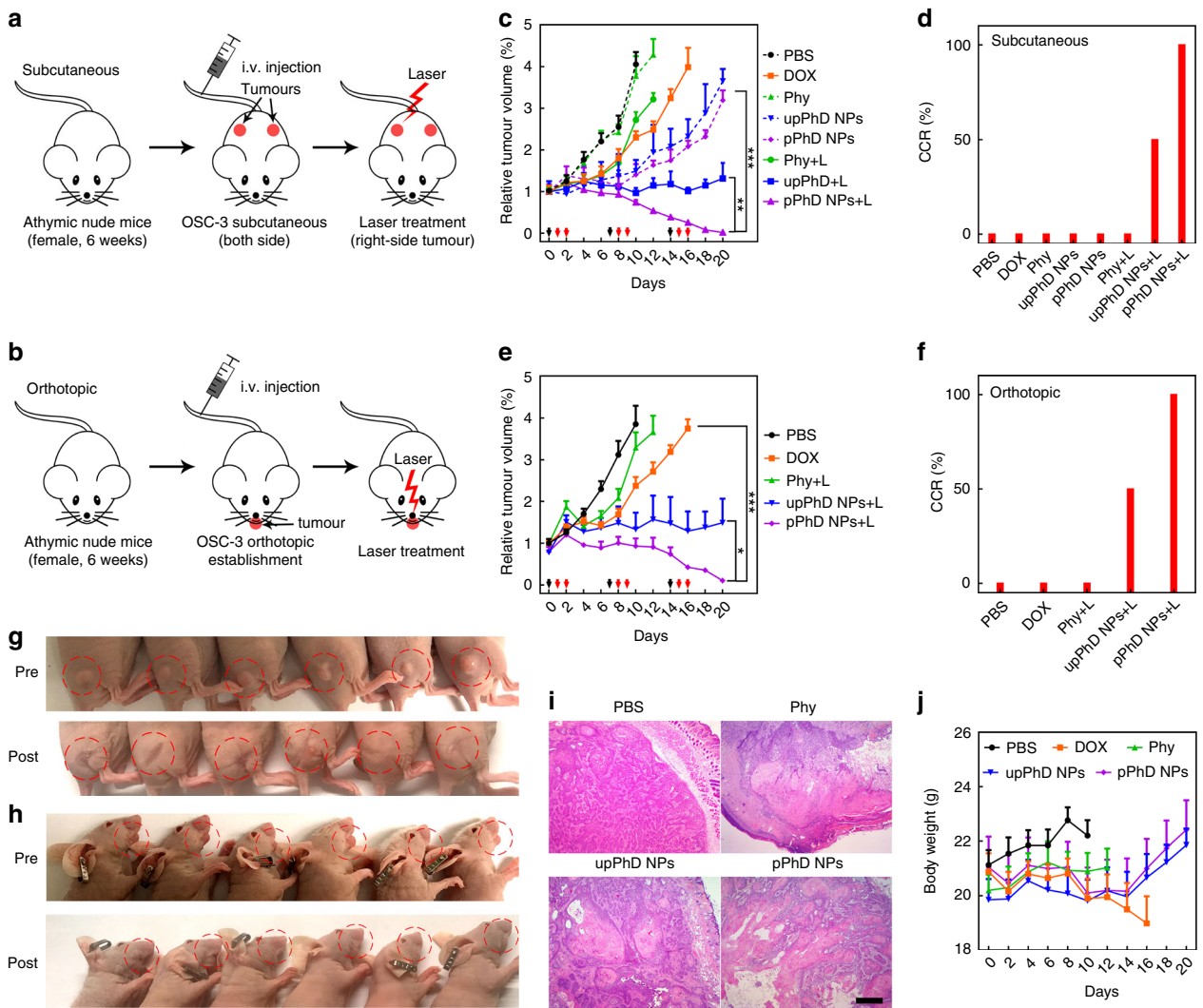

**Fig. 5** Therapeutic effects of the nanoparticles. The establishment of **a** subcutaneous and **b** orthotopic oral tumour models ($n = 6$), and the subsequent treatments with PBS, 4.7 mg kg$^{-1}$ DOX, 5.3 mg kg$^{-1}$ Phy, 10 mg kg$^{-1}$ upPhD NPs and 10 mg kg$^{-1}$ pPhD NPs (calculated based on the concentration of PhD monomer), respectively. The doses of free DOX and Phy were equivalent to those in upPhD NPs and pPhD NPs groups, respectively. The laser (680 nm) doses were all set as 0.4 W cm$^{-2}$ for 3 min. **c** The tumour volume changes on subcutaneous tumours ($n = 6$) after administration of various treatment groups. The black arrows denote the nanoparticles administration, red ones point out the tumour treated by laser treatments. **d** The complete cure rate (CCR%) of the subcutaneous tumours. **e** The tumour volume changes on orthotopic tumours ($n = 6$). **f** The CCR% of the orthotopic tumours treated with different groups. The images showed the tumour profiles of **g** the subcutaneous and **h** the orthotopic models, before (Pre) and after (Post) the pPhD NPs plus laser treatments. The mice were randomly aligned, the upper and lower panel may not correspond to each other. **i** H&E showed the histological changes induced by the in vivo phototherapeutic effect of photosensitizer-harboured materials. PBS group treated with laser was employed as control. The scale bar is 200 μm. **j** Body weights changes ($n = 6$) of tumour-bearing mice after treatment. \*$p < 0.05$; \*\*$p < 0.01$; \*\*\*$p < 0.001$. All error bars are presented as standard deviation. Note: the mice were immunodeficient, the laser treatments on right tumour were not able to induce immuno-responses to affect the left tumour

tumours within 2 weeks. The free photosensitizer with laser (Phy + L) and free chemo-drug (DOX) exhibited moderate anti-tumour activity but could not slow down the tumour growth. The nanoformulation groups without laser (upPhD NPs and pPhD NPs) showed considerable anti-tumour efficacy than free chemo-drugs, suggesting that our nanoparticles could improve drug efficacy. The group of upPhD NPs with laser (upPhD NPs + L) showed more effective tumour inhibition and effectively prevented the tumour progress. Most interestingly, the PEGylated nanoparticles (pPhD NPs + L) exhibited exceptional anti-tumour efficacy, with 100% complete cure rate (Fig. 5d), which was much higher than the laser-treated un-PEGylated nanoparticles (50%), and other control groups (0%). The best anti-tumour efficacy of

pPhD NPs + L group was further demonstrated in orthotopic tumours (Fig. 5e), which achieved 100% complete cure rate as well (Fig. 5f). The tumour images of subcutaneous and orthotopic models that treated by pPhD NPs + L (Fig. 5g, h) and upPhD NPs + L (Supplementary Figure 22 & 23) further indicated the superiority of the PEGylation. H&E staining was utilised to evaluate the phototherapeutic effect of free photosensitizer (Phy), un-PEGylated nanoparticles (upPhD NPs) and PEGylated nanoparticles (pPhD NPs) in tumour tissue compared with the PBS group. As shown in Fig. 5i, all the phototherapy groups caused different extent of tumour tissue damage, such as cellular destruction and necrosis, in which the pPhD NPs induced the largest areas of damage in treated tumour tissue. Fig. 5j showed

the body weights changes of the mice along the duration of the treatments. DOX-induced obvious body weights loss after the second dose of treatment; pPhD NPs did not exhibit systemic toxicity, since the mice gained body weights during the treatment.

**Further investigation of transformability and functionalities.** We then introduced more control groups, such as non-transformable pPhD NPs (NT-pPhD NPs), nanoformulation of DOX (Doxil) and pPhD NPs with low laser dose, to further investigate the dual-transformability, chemotherapeutic delivery and photodynamic therapy alone of pPhD NPs, respectively. The non-transformable pPhD NPs (NT-pPhD NPs) were developed by introducing non-cleavable chemical bonds between the PEG and nanoparticles as a more appropriate non-transformable control. The NT-pPhD NPs were fabricated by following the same procedures for the preparation of pPhD NPs, except that the PEG-2CHO was replaced by PEG-2COOH NHS ester (Supplementary Figure 24). The PEG-2COOH NHS ester reacted with the amine groups on the surface of upPhD NPs and formed non-cleavable amide bonds. Such PEGylation could not be detached, and thus the resulting nanoparticles were not transformable. The morphology and size distribution of NT-pPhD NPs were shown in Fig. 6a, NT-pPhD NPs presented similar morphology as pPhD NPs and exhibited similar hydrodynamic size at ~87 nm. The dual-transformability was also investigated (Fig. 6b), neither the size nor the surface charge was changed after being stimulated by pHe (pH 6.8). The building blocks, morphology, surface modification, size distribution and surface charge of NT-pPhD NPs were very similar to these of pPhD NPs, indicating that the NT-pPhD NPs were an ideal control to elucidate the importance of dual-transformability in vivo. After the NT-pPhD NPs were prepared, a new set of animal studies were conducted to evaluate the anti-tumour efficacy of the NT-pPhD NPs, Doxil and pPhD NPs (with lower laser dose) in subcutaneous and orthotopic oral cancer models. Followed the same protocol that used in Fig. 5, the mice were randomly assigned into 5 groups ($n = 6$): PBS, Doxil, NT-pPhD NPs, pPhD NPs and pPhD NPs with low laser dose. The low laser dose was set to 0.2 W cm$^{-2}$ to generate PDT dominant effect as reported previously[39]. The total laser power was kept at the same level for the low dose (0.2 W cm$^{-2}$, 6 min) and high dose groups (0.4 W cm$^{-2}$, 3 min). In subcutaneous model (Fig. 6c), all the tumours in PBS group grew fast and the mice died within two weeks. The Doxil exhibited excellent anti-tumour activity and could effectively slow down the tumour growth. The efficacy of chemotherapeutic function alone of pPhD NPs without laser irradiation was comparable to that of Doxil (no statistically significant difference) and such monotherapy could effectively slow down the tumour progress. The effectiveness of pPhD NPs can be ascribed to their unique dual-transformability. Although they were very similar to pPhD NPs except the transformability, NT-pPhD NPs only showed marginal efficacy, which was ascribed to their limited penetration in tumours. We further conducted microscopic imaging studies at tissue level to support this claim. As shown in Fig. 6g, NT-pPhD NPs only stayed in the periphery of the tumour blood vessels. In comparison, the pPhD NPs showed superior tumour penetration and lit up the tissue throughout the tumour. The anti-tumour efficacy and tissue penetration combined to support that the transformability of nanoparticles greatly contributed to the efficacy. The photo-therapy could be introduced to further enhance the efficacy by applying laser irradiation on the tumours. The laser treatment extensively enhanced the anti-tumour efficacy of NT-pPhD NPs, and the tumour shrunk to smaller size, and one mouse in this group was completely cured (Fig. 6d). The pPhD NPs showed exceptional efficacy, and all the tumours were completely ablated

(CCR is 100%), which was dramatically more effective than NT-pPhD NPs (Fig. 6d). The remarkable improvements of laser-induced anti-tumour efficacy were ascribed to the extremely effective PTT. As shown in Supplementary Figure 25, the laser treatment elevated the temperature for more than ~17 °C in NT-pPhD NPs and ~22 °C in pPhD NPs treated tumours, such feverish temperatures were strong enough to ablate the tumour tissue. Since the PTT at high laser dose was extremely powerful which could completely deluge the PDT and chemotherapy, we introduced a low laser dose treatment (0.2 W cm$^{-2}$, 6 min) to pPhD NPs to evaluate the combination of the chemo- and pho-todynamic- therapy. Such low laser dose only led to similar photothermal effect to PBS group with 0.4 W cm$^{-2}$ laser (Supplementary Figure 25), which yielded very limited efficacy on the tumours (PBS + L in Fig. 6c). Due to the presence of photo-sensitizer, this low laser dose still elicited abundant ROS production for PDT (Supplementary Figure 26). As shown in Fig. 6c, the PDT alone enabled to extensively improve the efficacy of pPhD NPs by comparing with the non-laser-treated pPhD NPs and Doxil groups, supporting that PDT played a significant role in the anti-tumour activities. In the orthotopic oral cancer model, each group showed similar efficacy as that in subcutaneous model (Fig. 6e). Doxil effectively slowed down the tumour progress; NT-pPhD NPs showed excellent anti-tumour efficacy due to the strong PTT effect; and pPhD NPs with high laser dose exhibited the best anti-tumour efficacy and achieved 100% CCR (Fig. 6f). The PDT and chemotherapy of pPhD NPs (0.2 W cm$^{-2}$, 6 min) resulted in excellent anti-tumour efficacy, which were much effective than Doxil. With the introduction of control groups, such as NT-pPhD NPs, Doxil and pPhD NPs (0.2), we systemically elucidated the benefits of the dual-transformability, chemotherapeutic delivery and role of the PDT alone for tumour treatments of the pPhD NPs. The body weight for all treatment group (Fig. 6h) was similar to that of PBS group, indicating that these treatment groups exhibited no obvious toxicity in mice.

**Systemic toxicity evaluations of pPhD NPs.** The in vivo toxicity to the main organs was evaluated by monitoring hematoxylin & eosin (H&E) staining. The lesion of major organs was evaluated by H&E staining (Supplementary Figure 27). DOX showed obvious liver and heart toxicity, the striated muscle of heart disappeared. All other groups didn't exhibit distinguishable abnormality, indicated that our nanoformulation could extensively decrease the systemic toxicities of chemotherapeutic drugs.

## Discussion
In this work, we developed dual-transformable, Trojan-Horse nanoparticles (pPhD NPs), which enabled efficient delivery of ultra-small, full API and multifunctional nanotheranostics for cancer imaging and therapy. This nanoplatform was demonstrated to have integrated highly efficient drug delivery functions as well as synergistic trimodality therapeutic and dual-modal imaging functions in one simple formulation. The pPhD NPs possessed larger size and slightly positive charge with intraparticle cross-linkages under normal physiological conditions and could be responsively transformed to ultra-small nanotheranostics with strongly positive charge in TME. The initial state of the nano-particles with PEG surface was beneficial to the drug delivery in blood vessel. Both post-transformed and dynamically trans-formed nanoparticles enabled to improve the tumour cell uptake and tumour interstitial penetrations, because of the strongly positive charge and the ultra-small particle size. These features endowed our nanoparticles with excellent capability to system-atically circumvent the sequential biological barriers which had hindered the drug delivery. We directly observed the size

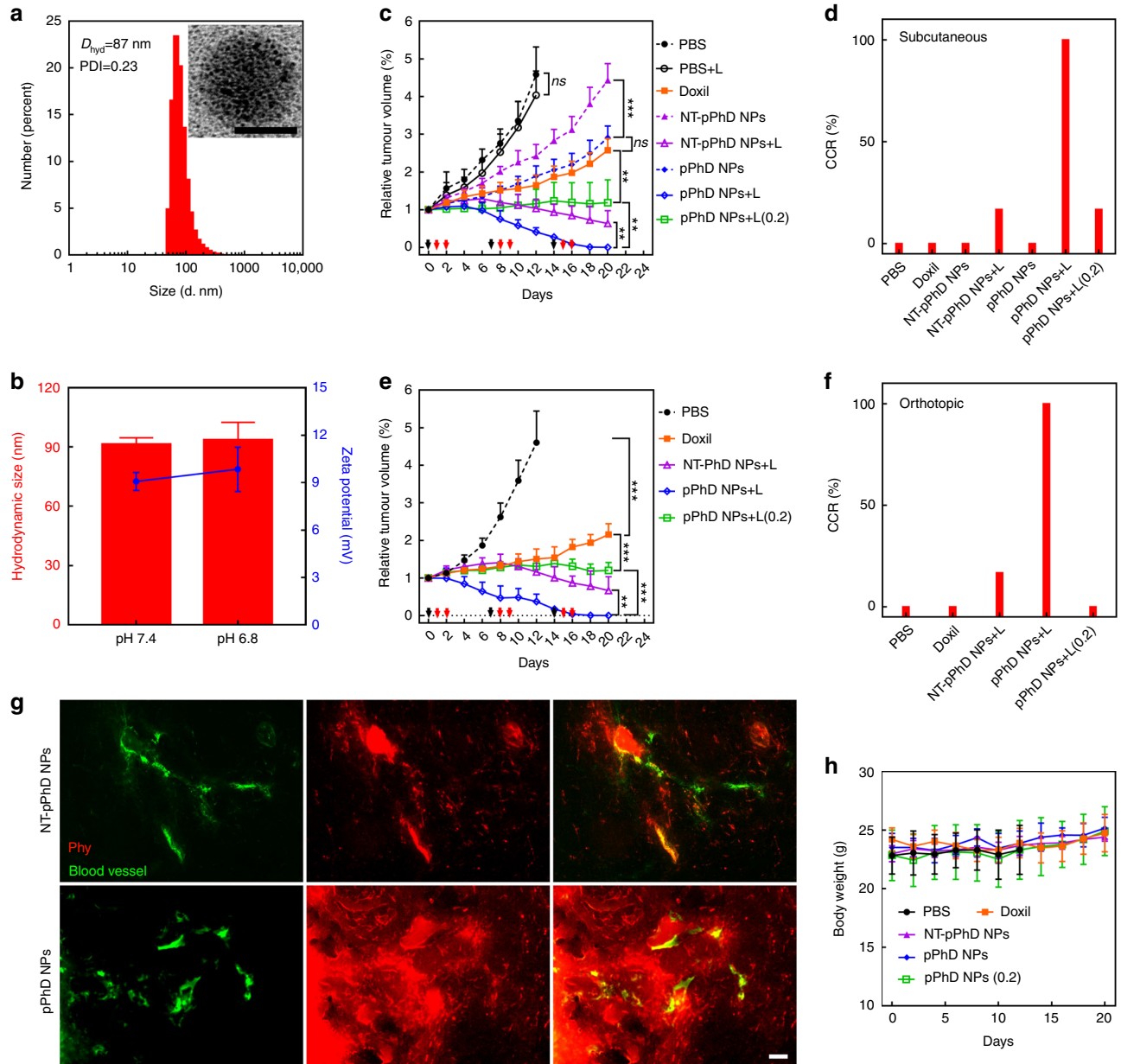

**Fig. 6** In vivo investigation of transformability, chemotherapeutic delivery and photodynamic effect. **a** Size distribution and morphology of NT-pPhD NPs. The inset is TEM micrograph with a scale bar of 50 nm. **b** Size and surface charge changes of NT-pPhD NPs before and after the stimulation at pHe (6.8). **c** The tumour volume changes on subcutaneous tumours ($n = 6$) after administration of various treatment groups. The black arrows denote the nanoparticles administration, and red ones point out the tumour treated by laser treatments. **d** The complete cure rate (CCR%) of the subcutaneous tumours. **e** The tumour volume changes on orthotopic tumours ($n = 6$). **f** The CCR% of the orthotopic tumours. **g** The microscopic distribution of transformable nanoparticles (pPhD NPs) in tumour tissue in comparison with the non-transformable pPhD NPs (NT-pPhD NPs) to prove the superior tumour penetrations. The blood vessel (green) was stained by FITC-dextran (70k) while the red fluorescence indicated the distribution of nanoparticles. The scale bar is 50 μm. **h** Body weights changes of tumour-bearing mice after treatment ($n = 6$). The treatment doses were set as 4.7 mg kg$^{-1}$ for Doxil (calculated by DOX concentration), 5.3 mg kg$^{-1}$ for Phy, 10 mg/kg for NT-pPhD NPs and 10 mg kg$^{-1}$ for pPhD NPs (NT-pPhD NPs and pPhD NPs both were calculated based on the concentration of PhD monomer), respectively. For the group of treated with pPhD NPs at low laser dose, the laser (680 nm) dose was set to 0.2 W cm$^{-2}$ for 6 min. The other laser-treated groups were all set as 0.4 W cm$^{-2}$ for 3 min. *$p < 0.05$; **$p < 0.01$; ***$p < 0.001$; ns, not significant. All error bars are presented as standard deviation. Note: the mice were immunodeficient, the laser treatments on right tumour were not able to induce immuno-responses to affect the left tumour

transformability by TEM, and validated the elevated cell uptake, enhanced tumour accumulation and superior penetration of pPhD NPs. The in vivo therapeutic studies supported that the PEGylated and transformable nanoparticles could greatly improve the therapeutic effect over the non-PEGylated and non-transformable nanoparticles. Furthermore, the versatile theranostic functions of pPhD NPs are highly valuable for cancer

treatment. In comparison with the commercially available and well-developed Doxil, pPhD NPs showed comparable chemotherapeutic effect as a monotherapy approach in OSC-3 oral cancer model. When the PDT was introduced by using low dose of laser, pPhD NPs were more efficacious than Doxil. When photothermal therapy was involved by using high dose of laser, the efficacy of pPhD NPs overwhelmed all other treatments,

including Doxil. In comparison with other reported non-PEGylated full API nanoparticles thus far[21,40–42], pPhD NPs yielded 100% CCR and showed better anti-tumour efficacy, due to their excellent multimodal therapeutic functions, PEGylated cross-linkage and dual-transformability. By comparing with the multimodal full API nanoparticles that we developed, such as upPhD NPs in this work and PaIr NPs[43], the anti-tumour efficacy of pPhD NPs was also superior, which can be ascribed to their excellent dual-transformability and PEGylated cross-linkage. pPhD NPs could be conveniently prepared by simply using bio-compatible PEG as the only excipient, the DOX and Pa as the APIs. We utilised simple materials, but integrated versatile functionalities in one single smart nanoplatform. With the simple introduction of PEG, we realised dual-transformability, which could sophisticatedly overcome the drug delivery barriers and largely enhance the anti-tumour efficacy. The dual-transformable, Trojan-Horse nanoparticles integrated trimodality therapeutic functions in a single formulation, including photothermal-, photodynamic- and chemotherapies. These treatment modalities are complementary and synergistic. The phototherapy provides an immediate, extensive tumour abrogation, and is highly useful to treat chemo-resistant tumours[44,45]. While the chemotherapy can provide sustainable therapeutic effect and is suitable to treat tumours that are beyond the reach of the light. In the in vivo therapeutic studies, the pPhD NPs with laser-treated tumour were completely cured in both orthotropic and subcutaneous models. Furthermore, our nanoparticles were intrinsically useful for dual-modal imaging (NIRFI/MRI) and realised the nanoscale integration of imaging and therapeutic functions. The in vivo therapeutic delivery, the time-dependent tumour accumulation and the tumour response could be conveniently visualised by these imaging modalities. These highly integrated theranostic functions are very useful for individualised adjustment of treatment toward precision nanomedicine.

This transformable, Trojan-Horse liked nanoplatform with integrated powerful delivery efficiency and versatile theranostic functions are expected to open a broad range of possibilities to improve cancer management. It will also inspire other scientists to develop next generation smart nanomaterials for a variety of biomedical applications.

## Methods

**Materials and instruments**. Pheophorbide a was bought from Santa Cruz Biotechnology (TX, USA). Doxorubicin was purchased from LC Laboratories (MA, USA). Doxil (2 mg mL$^{-1}$) was purchased from Janssen Pharmaceutica (NV, USA). Hydrazine, (1-ethyl-3-(3-dimethyl aminopropyl) carbodiimide hydrochloride) (EDC), N-hydroxysuccinimide (NHS), N, N'-Dicyclohexylcarbodiimide (DCC), 4-Dimethylaminopyridine (DMAP), 2′,7′-Dichlorofluorescin diacetate (DCF-DA), MnCl$_2$ and all solvents were purchased from Sigma-Aldrich (MO, USA). Singlet oxygen sensor green (SOSG), LysoTracker Deep Red and CellROX were purchased from Thermo Fisher Scientific Inc. The synthetic compounds were analysed by Bruker UltraFlextreme matrix-assisted laser desorption/ionisation time of flight mass spectrometry (MALDI-TOF-MS), Thermo Electron LTQ-Orbitrap XL Hybrid electrospray ionisation mass spectrometry (ESI-MS) and 600 MHz Avance III nuclear magnetic resonance (NMR) spectrometer (Bruker, German). Transmission electron microscopy (TEM) was performed on a Talos L120C TEM (FEI) with 80 kV acceleration voltage. The in vitro laser treatments were conducted under light source that with broader covering area (Omnilux new-U). Cell fluorescence images were captured with a confocal laser scanning microscopy (CLSM, LSM810, Carl Zeiss). The magnetic resonance imaging (MRI) was conducted on a Biospec 7 T MRI scanner (Bruker, German). Apoptosis and cell ROS production were evaluated by a BD Fortessa 20 colour flow cytometry. Hydroxylated Polyethylene glycol 2000 (PEG$_{2000}$) was purchased from Laysan Bio Inc (AL, USA).

**Synthesis of pheophorbide a-hydrazide (Phy)**. 594 mg pheophorbide a (~1 mmol), 383 mg EDC (2 mmol) and 230 mg NHS (2 mmol) were dissolved in 20 mL dichloromethane (DCM), and vigorously stirred at room temperature for 30 min. Then 188 µL anhydrous hydrazine (6 mmol) was added into the reaction system. The reaction was under vigorously stirring at room temperature for another 4 h. Then, the pheophorbide a-hydrazide (Phy) was extracted from the reaction system with DCM against water. The yield was ~83%. HRMS: m/z M + [H]$^+$ was

607.3010 Da. $^1$H-NMR (DMSO-d6, 600 MHz): δ 9.69 (s, 1H), 9.33 (s, 1H), 8.90 (s, 1H), 8.16 (dd, J$_1$ = 17.4 Hz, J$_2$ = 11.4 Hz, 1H), 6.42 (s, 1H), 6.37 (d, J = 18.0 Hz, 1H), 6.37 (d, J = 11.4 Hz, 1H), 4.58 (m, 1H), 4.05 (m, 1H), 3.84 (s, 3H), 3.80 (s, 1H), 3.63 (m, 6H), 3.41 (s, 3H), 3.12 (s, 3H), 2.25 (m, 2H), 2.00 (m, 1H), 1.80 (d, J = 7.8 Hz, 2H), 1.60 (m, 4H).

**Synthesis of pheophorbide a-hydrazide-doxorubicin (PhD)**. 121.4 mg Phy (0.2 mmol) and 58 mg doxorubicin hydrochloride (0.1 mmol) with a drop of TFA (20 µL) were dissolved in 10 mL methanol and stirred overnight under 50 °C. The target compound (PhD) was purified by column chromatography. The yield was ~42%. HRMS: m/z M + [H]$^+$ was 1132.4704 Da. $^1$H-NMR (DMSO-d6, 600 MHz): δ 10.26 (s, 1H), 9.63 (s, 1H), 9.20 (s, 1H), 8.16 (s, 1H), 8.01 (dd, J$_1$ = 18.0 Hz, J$_2$ = 12.0 Hz, 1H), 7.38 (m, 2H), 7.13 (d, J = 6.6 Hz, 1H), 6.28 (t, J = 12.6 Hz, 1H), 6.04 (d, J = 10.8 Hz, 1H), 5.76 (s, 2H), 5.66 (s, 1H), 5.22 (s, 1H), 4.90 (s, 1H), 4.50 (m, 2H), 3.99 (s, 2H), 3.87 (m, 2H), 3.77 (s, 3H), 3.69 (s, 3H), 3.62 (s, 3H), 3.58 (m. 3H), 3.31 (m, 6H), 3.18 (m, 1H), 3.06 (s, 2H), 2.55 (s, 1H), 1.91 (m, 1H), 1.72 (d, J = 7.8 Hz, 2H), 1.61 (t, J = 7.8 Hz, 3H), 1.2 (m, 5H), 0.9 (m, 4H).

**Synthesis and characterisation of dual-aldehyde terminated PEG**. 570 mg 4-Formylbenonic acid (5 mmol) and 206 mg DCC (7 mmol) were dissolved in anhydrous (DCM). The mixture was stirred at 0 °C for 30 min until plenty of white precipitates were observed. Then, 1000 mg hydroxylated PEG$_{2000}$ (0.5 mmol) and 73 mg DMAP (0.6 mmol) in 10 mL anhydrous DCM was added. The resulting mixture was stirred at ambient temperature for 24 h. The dual-aldehyde terminated PEG was purified by precipitation via cold ether, and further dialysed with a dialysis tube (MWCO is 1000 Da). The solution was then lyophilised. The MALDI-TOF MS showed obvious spectra shift after dual-aldehyde was introduced. The yield was ~87%. $^1$H-NMR (DMSO-d6, 600 MHz): δ 10.13 (s, 2H), 8.17 (d, J = 0.84, 4H), 8.07 (d, J = 0.84, 4H), 4.45 (m, 4H), 3.79 (m, 4H), 3.63 (m, 4H), 3.55 (m, 4H), 3.51 (m, 164H).

**Preparation and characterisation of PEGylated PhD NPs (pPhD NPs)**. The nanoparticles were prepared by the following typical re-precipitation method[25,46]. Briefly, 50 mmol PhD DMSO solution was firstly made, and 2 µL PhD solution was then dropped into 998 µL Milli Q water under sonication, followed by a 3~5 s vortex, resulting in the unPEGylated PhD NPs. Then, 100 µM dual-aldehyde terminated PEG were added, and stirred under ambient temperature for 48 h, resulting in the the pPhD NPs. The size distributions, PDI and surface charge of the nanoparticles were carried on with a DLS (Zetasizer, Nano ZS) from Malvern Instruments Ltd (Worcestershire, UK). The morphology of NPs was observed by a Talos L120C TEM (FEI) with 80 kV acceleration voltage. The TEM samples were prepared by dropping aqueous nanoparticle solution (50 µM) on copper grids and naturally dried under room temperature.

**Mn$^{2+}$ chelation of pPhD NPs**. Mn$^{2+}$ chelation was conducted by following the published method[37]. In brief, 24.3 mg Phy (40 µmol) with five times of MnCl$_2$ (25.2 mg, 200 µmol) were dissolved in 2 mL anhydrous methanol, then 200 µL pyridine was added. The reaction system was under vigorous agitation and refluxed for 2 h. The chelated Phy was purified by funnel separation (DCM against water) for five times. The un-chelated Mn$^{2+}$ dissolved in Milli Q water and was removed. The Mn$^{2+}$ chelated Phy distributed in organic layer (DCM), and was dried with a rotavapor. Then, the manganese ion chelated Phy was employed to synthesise PhD monomers and fabricate the pPhD NPs by following the procedures mentioned above.

**Optical measurements**. The UV–vis spectra were collected with a UV–vis spectrometer (UV-1800, Shimadzu). For all materials and compounds, the absorbance was collected under a range of 200–800 nm. The fluorescence spectra were obtained by a fluorescence spectrometer (RF-6000, Shimadzu). The excitation and emission bandwidths were both set as 5.0 nm, and the data interval was 1 nm. For Phy, the excitation of 412 nm was used, while the emissive band was scanned from 432 to 800 nm. For DOX, the excitation was 488 nm, while the emission scanning ranged from 508 to 800 nm. To measure the fluorescence properties of PhD monomer and nanoformulation, both excitations were employed. For optical measurements, the quartz cuvettes were with 1 mm path length.

**Stability of pPhD NPs in serum**. 100 µL pPhD NPs were incubated in 10% foetal bovine serum and kept in a cell culture incubator at 37 °C. The hydrodynamic size of the nanoparticles was continuously monitored by DLS.

**CAC of pPhD NPs**. Pyrene molecules were employed as an indicator to determine the CAC of nanoparticles by comparing the fluorescence of their third and the first emissive peaks (I$_3$/I$_1$). First, pPhD NPs were diluted into different concentrations (0.01, 0.05, 0.1, 0.5, 1, 5, 10 and 50 µM); then, 999 µL pPhD NPs of each dilution was incubated with 1 µL pyrene acetone solution (0.1 mM) at 37 °C for 2 h. The fluorescence spectra of pyrene (excitation is 335 nm) in different pPhD NPs dilutions were recorded. The fluorescence intensity ratio (I$_3$/I$_1$) of the third and first emissive peaks were measured for CAC calculation.

**Near-infrared fluorescence imaging (NIRFI) of pPhD NPs**. 10 μL PhD monomers and pPhD NPs with varied concentrations were dropped on a transparent film respectively, and put in the NIRFI chamber, and their NIRFI was collected by using a Kodak multimodal imaging system IS2000MM with an excitation at 625 ± 20 nm and an emission at 700 ± 35 nm. The PhD monomers were obtained by dissolve PhD molecules in good solvent (DMSO).

**Photothermal and photodynamic effects of pPhD NPs**. For photothermal effect evaluation, pPhD NPs with different concentrations (1, 5, 10 and 50 μM) were placed in 96-well plate, and treated with 0.4 W cm$^{-2}$ laser (680 nm) for 3 min. Water solution was set as control (0 μM), and treated with identical laser exposure. The hyperthermia produced by pPhD NPs were monitored by a FLIR thermal camera. The photodynamic effect of pPhD NPs was indicated by the level of laser-triggered ROS. The ROS production was visualised by a commercial probe, singlet oxygen sensor green (SOSG). Similar to photothermal effect analysis, different concentrations of pPhD NPs were incubated with SOSG probe, and exposed under laser (680 nm, 0.4 W cm$^{-2}$) for 3 min. SOSG probe incubated with water was set as blank control by treated with same laser dose. The fluorescence readouts of SOSG was monitored by a microplate reader (SpectraMax M2, Molecular Devices) to indicate the photodynamic effect that elicited by photosensitizer-harboured pPhD NPs.

**LC-MS analysis of the cleavage of hydrazone bond in PhD monomer**. The triple quadrupole LC-MS/MS system consisted of a 1200 series HPLC system (Agilent Technologies, USA) and a mass spectrometer (6420 triple Quad LC/MS, Agilent Technologies, USA). Chromatographic separation was carried out on a Waters XBridge-C$_{18}$ (2.1 mm × 50 mm, 3.5 μm) column at 40 °C with 10 mM ammonium acetate 0.1% formic acid aqueous solution mobile as phase A and acetonitrile as mobile phase B. 10 μL, 100 μM pre-treated PhD monomers (at pH 5.0 for overnight) were injected into C$_{18}$ column. The gradient was gradually changed from 10% mobile phase B (90% phase A) to 90% mobile phase B (10% phase A) within 9 min. The MS parameters were as follows: capillary, 5000 V; gas temperature, 320 °C; gas flow, 8 mL per minute; and nebuliser, 40 psi. The data analysis was performed by Mass Hunter Workstation Software Qualitative Analysis (Version B.06.00) and Quantitative Analysis (Version B.05.02).

**Accumulated drug release of pPhD NPs triggered by acidic pH and laser**. 100 μM pPhD NPs were prepared and loaded into dialysis cartridges (MWCO is 3,500 Da) to determine the accumulated drug release profile. The cartridges were submerged into 1000 mL PBS (pH 7.4) and acidized PBS (pH 5.0) respectively, and stirred with a moderate-speed at ambient temperature. The laser-triggered drug release was conducted by irradiation with laser at 0.4 W cm$^{-2}$ for 3 min before the dialysis. The DOX remained in the dialysis cartridge was drawn with a micro-syringe at various time points, and quantitatively measured by the UV–vis absorbance of DOX. Each value was reported as the means of the triplicate samples.

**Cell line**. The human oral squamous cell carcinoma cells (OSC-3) line was kindly courtesied from Dr. Kit S. Lam's lab. Cell line authentication was performed by short tandem repeat DNA profiling. The cell line has been tested for mycoplasma contamination routinely. Cell culture was performed in a cell incubator (5% carbon dioxide and 10% humidity. Temperature is 37 °C). The cells were cultured in RPMI-1640 medium with 10% FBS and antibiotics containing penicillin and streptomycin.

**Cell viability assay**. Human oral squamous cell carcinoma cells (OSC-3) were seeded in 96-well plate with a density of 6000 cells per well. The cells were incubated overnight until fully attached. Then the cells were treated with different concentrations of various drug formulations. After 12 h treatments, the cells were washed, and fresh medium was added. For the light treated groups, the cells were exposed to the light for 3 min, and further incubated for another 24 h in parallel with non-light treated groups. The cell viability was measured by MTT assay. Results were shown in form of average cell viability [(OD$_{treat}$-OD$_{blank}$)/(OD$_{control}$-OD$_{blank}$)×100%] of triplicate wells.

**Cell uptake assay**. Since the fluorescence of Phy and DOX were both quenched in pPhD NPs, the cell uptake of pPhD NPs may not be accurately measured if the measurements were conducted under the aqueous circumstance. To evaluate the cell internalisation of pPhD NPs and their post-transformed counterparts (pre-treated with pH 6.8 to achieve size/charge dual-transformability). pPhD NPs and post-transformed pPhD NPs were incubated with OSC-3 cells for 3 h, respectively, and then the cells were detached and collected in a vial. After removing the medium, the OSC-3 cells were dissolved with the same volume of DMSO to dissolve the cells and completely dissolve all materials that related to pPhD NPs. The solutions were then evaluated by fluorescence spectrometer to test the fluorescence of DOX. The DOX solution concentrations represent the cellular uptake of the pPhD NPs. Each value was reported as the means of the triplicate samples.

**ROS assay in cellular level**. OSC-3 cells were seeded in 6-well plates with $5.0 \times 10^5$ cells per well, and cultured for 24 h until fully attached. The cells were treated with Phy, pPhD (pH 7.4) and pPhD (pH 6.8) for 3 h. The cells were then incubated with DCF-DA (10 μM) for another 30-min followed by light treatment for 1 min and analysis by flow cytometry (BD FACSCanto™ flow cytometry system). Cells without any treatment were used as a control. The concentrations of all materials were set as 10 μM.

**Apoptosis assay**. OSC-3 cells were seeded in 6-well plates with $5.0 \times 10^5$ cells per well, and cultured for 24 h until all cells were fully attached. The cells were treated with DOX, Phy, pPhD (pH 7.4) and pPhD (pH 6.8) for 3 h, then applied for light treatment for 1 min. 24 h later, all cells were harvested and collected in EP tubes, and washed twice (1000 rpm, 5 min, 4 °C) with cold PBS and then suspended in 100 μL binding buffer. Then, 5 μL Annexin V-FITC (20 μg/mL) was added to each tube and incubated for 15 min at 4 °C in dark. After gentle vortex, 2 μL PI (50 μg/mL) was added to each tube. The cells were filtered with Falcon™ Cell Strainers prior to the flow cytometry analysis. The cells without any stain were set as negative control. The concentrations of all materials were set as 10 μM.

**Lysosomes colocalization assay**. OSC-3 cells were incubated with 20 μM pPhD NPs for 4 h, then stained with Lysotracker Deep Red for confocal laser scanning microscopy (CLSM) observation. The fluorescence spectrum of Lysotracker Deep Red overlapped with that of Phy, but the fluorescence readouts were much higher than Phy under Cy5 channel. We, therefore, adjusted the parameters of CLSM until we could not observe the fluorescence of Phy in pPhD NPs treated cells, and used these parameters to observe the fluorescence of Lysotracker Deep Red to avoid the interference of Phy. For DOX distribution, standard FITC channel was used.

**Cell spheroids penetration of the nanoparticles**. OSC-3 cells were seeded in the round-shape bottom 96-well plate at a density of $10^4$ cell per well. Cell spheroids were treated with 20 μM pPhD (pH 7.4) and 20 μM post-transformed pPhD (pH 6.8). The penetrations of the nanoparticles were monitored by a confocal laser scanning microscopy.

**Phototherapeutic effect on cells**. OSC-3 cells were seeded in 8-well chamber slide with $5.0 \times 10^4$ cells per well, and cultured for 24 h until all cells were completely attached. The cells were then treated with 10 μM pPhD NPs for 3 h. Cells without any treatment were used as a control. Both treatments were exposed to light for 1 min. After light treatments, the OSC-3 cells were stained with 40 nM of DiOC$_6$(3) for 20 min to indicate the live cells, and propidium iodide (PI) for 20 min to label the dead cells. Confocal laser scanning microscopy was employed to monitor the photocytotoxicity to cells.

**Pharmacokinetics evaluation**. The jugular vein of male Sprague–Dawley rats was cannulated, and a catheter was implanted for intravenous injection and blood collection (Harland, Indianapolis, IN, USA). pPhD NPs (10 mg kg$^{-1}$, calculated based on the concentration of PhD monomer), upPhD NPs (10 mg kg$^{-1}$) and free DOX (4.7 mg kg$^{-1}$) were i.v. administrated into rat ($n = 3$). The doses of free DOX were equivalent to those in upPhD NPs and pPhD NPs groups. Whole blood samples (~100 μL) were collected via jugular vein catheter before dosing and at predetermined time points post-injection. The whole blood samples were then centrifuged for serum collection, then the blood serum was diluted with DMSO (20 μL serum was added to 80 μL DMSO) for fluorescence measurements. The kinetics of free DOX was measured through testing the fluorescence of 591 nm (excitation is 488 nm). For nanoformulation, the concentrations were measured by testing the fluorescence of Phy (Ex = 412 nm and Em = 680 nm), because the fluorescence of DOX was slightly quenched when conjugated to Phy. The values were calculated by molar concentration first, then exchanged to mg per mL. The values were plotted versus time after the subtraction of blood background.

**Establishment of animal models and treatment schedule**. Female athymic nude mice (6 weeks old) were purchased from Harlan (Livermore, CA, USA). All animal experiments were strictly in compliance with the guidelines of Animal Use and Care Administrative Advisory Committee of University of California, Davis. The subcutaneous tumour models were established by inoculated OSC-3 cells ($5 \times 10^6$ cells per tumour) into both flanks of the nude mice. The orthotopic models were established by inoculating OSC-3 ($5 \times 10^6$ cells per mouse) to the lips of the mice. After the subcutaneous tumours reached about 100 mm$^3$ and orthotopic tumours reached about 50 mm$^3$, mice were divided into five groups ($n = 6$): control (PBS), free drug (DOX), free photosensitizer (Phy), un-PEGylated PhD NPs (upPhD NPs) and PEGylated PhD NPs (pPhD NPs). The mice received materials via i.v. injection through the tail vein. The dose of DOX was 4.7 mg kg$^{-1}$, Phy was 5.3 mg kg$^{-1}$, pPhD NPs were 10 mg kg$^{-1}$ (calculated based on the concentration of PhD monomer) and upPhD NPs were 10 mg kg$^{-1}$. The doses of free DOX and Phy were equivalent to those in upPhD NPs and pPhD NPs groups. In subcutaneous models, the right tumours were subjected to laser exposure (0.4 W cm$^{-2}$, 3 min), and the left-side tumours were not treated with laser (to evaluate the efficacy of chemotherapy). In the orthotopic models, all the tumours that treated with

photosensitizer, including Phy, upPhD NPs and pPhD NPs, were treated with laser ($0.4\,W\,cm^{-2}$, 3 min). The laser treatments were introduced twice, at 24 h and 48 h after the i.v. injection. During the laser treatments, the photothermal effects were monitored and recorded by a FLIR infrared camera (FLIR Systems, Boston, MA).

**In vivo ROS production**. Orthotopic tumour-bearing mice were assigned into four groups ($n = 3$): 1) PBS, 2) Phy, 3) upPhD NPs and 4) pPhD NPs. $5.3\,mg\,kg^{-1}$ Phy, $10\,mg\,kg^{-1}$ upPhD NPs and $10\,mg\,kg^{-1}$ pPhD NPs (calculated based on the concentration of PhD monomer) were i.v. administrated into mice respectively. 24 h later, tumours of the mice were irradiated with $0.4\,W\,cm^{-2}$ laser for 3 min. The mice were sacrificed and the tumour was collected for NIRFI (Pre-cellROX). After the NIRFI, the tumours were immediately sunk into ROS probe solution (Cell-ROX) for 10 s, and conducted for another NIRFI (Post-cellROX). The in vivo ROS production was presented by fluorescence intensities of Post-cellROX deducted the fluorescence in Pre-cellROX tumours. The Phy signals were overlapped with cellROX, we deducted the NIRF of Phy (Pre-cellROX) from the final imaging results (Post-cellROX) to determine the ROS production.

**Biodistribution of the nanoparticles**. $10\,mg\,kg^{-1}$ upPhD NPs and $10\,mg\,kg^{-1}$ pPhD NPs (calculated based on the concentration of PhD monomer) were i.v. administrated to orthotopic tumour-bearing mice respectively. The tumours were then exposed to the laser at 24 h after the materials treatments. After the laser trigger, whole body imaging was acquired at indicated time points. After in vivo imaging, animals were sacrificed, and tumours and the major organs were harvested for ex vivo imaging.

**In vivo performance of pPhD NPs monitored by MRI**. For time-dependent tumour accumulation measurement, the orthotopic tumour models were i.v. injected with pPhD NPs ($10\,mg\,kg^{-1}$, calculated based on the concentration of PhD monomer. $Mn^{2+}$ dose: $0.01\,mmol\,kg^{-1}$), and the tumour area was monitor by a Bruker Biospec 7 T MRI scanner using T1-weighted Multi-Slice Multi Echo (MSME) sequence with echo time (TE) of 14 ms and repetition time (TR) of 500 ms. The matrix size is 256×256, and the field of view (FOV) is 3.50 cm. For monitoring the phototherapeutic effect, the OSC-3 tumour-bearing mice were treated with pPhD NPs (i.v. injection, $10\,mg\,kg^{-1}$), then the tumour site was exposed under continuous laser ($0.8\,W\,cm^{-2}$ for 3 min) at 24 h and 48 h after i.v. injection. The tumour conditions were monitored by MRI in real-time with the same parameters as that in tumour accumulation experiment. pPhD NPs doses were calculated based on the concentration of PhD monomer.

**Tissue penetration evaluation**. $10\,mg\,kg^{-1}$ NT-pPhD NPs and $10\,mg\,kg^{-1}$ pPhD NPs were i.v. injected into oral tumour-bearing mice, respectively. 24 h laser, 100 μL, $10\,mg\,mL^{-1}$ FITC-dextran (70 k) were i.v. injected to stain the blood vessel for 2 min. After that, the mice were sacrificed, the tumours were collected for cryo-section. Then, the tissue slices were subjected to CLSM observation. The thickness of slices was 20 μm. NT-pPhD NPs and pPhD NPs were both calculated based on the concentrations of PhD monomers, the amounts of PEG were excluded. For nanoparticles distribution, standard Cy 5 channel was used for observation of Phy; For blood vessel, standard GFP channel was used.

**Animal experiments for further investigation**. The subcutaneous and orthotopic oral cancer models were established by transplanting OSC-3 cells to two positions of the flanks or lips of nude mice, respectively. After tumour developed at 15 days, the mice were randomly assigned into different groups ($n = 6$) to receive the treatments. The groups included control (PBS), PEGylated PhD NPs (pPhD NPs), DOX nanoformulation (Doxil), non-transformable pPhD NPs (NT-pPhD NPs) and pPhD NPs with low laser dose groups. All tumour-bearing mice were treated once per week for 3 consecutive weeks by i.v. administration. In subcutaneous models (mice bearing two tumours), the right tumours that treated with photosensitizer-harboured materials were subjected to laser exposure ($0.4\,W\,cm^{-2}$, 3 min, the pPhD NPs with low laser dose group receive a dose of $0.2\,W\,cm^{-2}$ for 6 min), and the left-side tumours were not treated with laser to evaluate the efficacy of chemotherapy alone. In the orthotopic models, all tumours treated with photosensitizer-harboured materials were treated with laser. The laser treatments were given twice at 24 h and 48 h after the i.v. injection.

**Tumour volume and body weight measurements**. The body weights and tumour sizes were monitored three times a week, and the tumour volume was calculated by the following formula: Tumour volume = (Length × Width × Width)/2.

**H&E evaluation**. All laser-treated tumours were collected and stained with hematoxylin and eosin (H&E) to evaluate the effect of phototherapy. The main organs of each group, including heart, liver, spleen, lung, kidney, small intestine, were collected for H&E assay to evaluate the toxicity of the materials.

**Statistics**. Data statistics were analysed by calculating the t-test between two groups, and One-way ANOVA analysis of variations for multiple groups. Unless otherwise noted, all results were expressed as the mean ± s.d. A value of $p < 0.05$ was considered statistically significant.

## Data availability

All relevant data are included in the main manuscript and the Supplementary Information. Additional data are available from the corresponding author upon reasonable request.

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

## Acknowledgements

The authors thank the financial support from NIH/NCI (R01CA199668), NIH/NICHD (R01HD086195) and UC Davis Comprehensive Cancer Center Support Grant (CCSG) awarded by the National Cancer Institute (NCI P30CA093373).

## Author contributions

Y.L. and X.X. conceived the idea and designed the project. X.X. and Y.H. conducted all experiments and analysed the data. R.B., B.J. and W.Y. assisted with the animal studies. H.W., Y.Y. and Z.W. helped with the MRI studies. Z.M. assisted with the chemical synthesis and assigned the NMR spectra. D.J. helped with cytotoxicity assays. X.B.X. conducted the LC-MS analysis. T.L. assisted with the design and data analysis of all biological experiments. X.X. wrote the paper and all authors commented on the manuscript. Y.L. supervised the whole project.

## Additional information

**Competing interests:** Y.L., X.X., Y.H. and Z.M. are the co-inventors of the pending patent on the transformable nanoparticles and plan to develop the nanotherapeutics described in the manuscript commercially. The remaining authors declare no competing interests.

