## [Peer Review File · Nature Communications]

Reviewers' Comments:

Reviewer #1:

Remarks to the Author:

An interesting hierarchically-assembled nanoparticle, pPhD NP, was developed for dual modality tumor imaging and synergistic tumor therapy. The pPhD nanoplatform releases ultra-small therapeutic nanoparticles PhD in weakly acidic tumor microenvironment, then the ultra-small PhD releases pheophorbide a and doxorubicin. The pPhD nanoparticles not only showed tumor accumulation by near-infrared fluorescence imaging and magnetic resonance imaging, but also induced tumor eradication in different models, indicating the theranostic effect of the nanoplatform.

I recommend acceptance after addressing the minor issues:

1) Please provide more details about how the circulation half-lives of different NPs (pPhD and upPhD) are calculated based on Figure S15. The injected equivalent amount of DOX should be the same for all three treatment arms. Thus, the initial plasma concentration right after injection should be the same as well, correct?

2)"TEM micrograph (Figure 1b) demonstrated that the pPhD NPs were in spherical morphology, within which revealed a cluster of small dark dots. These small dots are believed to be a micellar assembly of PhD monomers, which further self-assembled into larger nanoaggregates through multi-micelle aggregation."

If the larger nanoaggregates are formed by micellular PhD monomers (suggested as small dots), should not we expect a uniform structure? Why does it show a heterogenous dotted structure?

Reviewer #2:

Remarks to the Author:

This manuscript reports study results of the development and evaluation of an interesting nanoparticle drug delivery system for the treatment of oral cancer. This dual size/charge-transformable nanoparticle has a drug delivery ultrasmall nanoparticle (4 nm) encapsulated in a PEG nanoparticle, resulting in a relatively small polymeric double layer nanoparticle carrying two therapeutic agents (pPhD NP). Extensive studies were done to fully characterize this drug delivery carrier. Result and figures are well-presented and support the conclusion. MRI contrast Mn²⁺-chelated pPhD NPs provided MRI capacity and in vivo results showed selective accumulation of pPhD into the tumors. Conditional release of ultra-small drug-nanoparticles from pPhD was also demonstrated. Furthermore, strong anti-tumor effect was found in the tumor bearing mice with the combination of delivery of Dox and photodynamic therapy. The design of the drug carrier is novel and based on special properties in tumor microenvironment to overcome drug delivery barrier.

Reviewer #3:

Remarks to the Author:

The strategy of using dual size/charge-transformable nanoparticles for drug delivery is well known so as the idea of using drug amphiphiles themselves to form nanoparticles. In this manuscript, authors created a pegylated pheophorbide a-hydrazone-doxorubicin nanoparticles (pPhD NP) by adapting these approaches with a newly synthesized, acid-labile porphyrin-doxorubicin conjugate, of which its porphyrin moiety presents both phototherapy and chemotherapy opportunities and multimodal imaging capabilities, which is also well established. So the novelty is modest. As for the concept, the overall design of pPhD NP for optimizing the ADMET process has merit albeit a very complex one. Hence the devils are in details in terms of fully supporting all the claims with rigorous experimental design and data. Unfortunately, the manuscript falls well short of this standard.

Some of the key concerns are:

1. Monomers (conjugates) characterization is not completed. Full characterization including the complete NMR assignments of any new compounds must be presented to meet the minimal ACS standard. There is also no molecular characterization for Phy-Mn²⁺.
2. Nanoparticle characterization is incompleting: a) despite the claim of ultra small size of 4nm for PhD NP in the abstract, there is no data to support it; b) Please provide the TEM of PhD NPs before and after the coating with PEG and explain the difference between PhD NPs and upPhD NPs (Fig S13 and Fig S14); c) based on the Figure 1a TEM data, size of pPhD NPs is smaller than 50nm, while its DLS showed 79nm with un-symmetric distribution, please explain; d) A cluster of small dark dots in pPhD NPs is not clearly visualized by the TEM image.
3. The methods used to measure absorption and fluorescence are not clearly described. It is surprised that the fluorescence (Figure 1d) was measured at such a high concentration (50uM) as fluorescence should be much more sensitive than absorption.
4. Missing of a critical control: upPhD NPs should not be called non-transformable nano-formulation as it has no PEGylation coating. A very important non-transformable control is a pH stable PEGylated pPhD NPs (e.g. using PEGylation with stable cross-linked chemistry instead of the formation of schiff base bond). This control should be used in drug release study (Figure 1F) and the further in vitro and in vivo studies, to prove nanoparticles transformation directed efficient drug delivery.
5. Pharmacokinetics measurement should be able to calculate blood circulation half-time to guide the in vivo study.
6. The biodistribution imaging data (Figure S16 and 17) are inconsistent to the ex. vivo biodistribution data in Figure 3 where the tumor uptake of the nanoparticle was far higher than the other organs.
7. In Figure 3H, the authors used Fluorescence imaging of ROS productions (NIRF ROS probe) to evaluate photodynamic (PDT) efficacy. Did author really believe that a potent PDT efficacy could be induced with light treatment at 0.4 w/cm² for 3 min.
8. Figure 3e displayed Mn-pPhD NPs based MRI. In Figure 3J, MRI was used to monitor phototreatment response (tumor shrinkage) for 168h. Was this MRI signal contributed by Mn-pPhD NPs based MRI (last 7days?) or agent free MRI?
9. Although a synergistic effect of pPhD NPs(photothermal-, photodynamic- and chemo-therapies) was claimed in the abstract, Figure 2i data only showed a combination effect.

Response to Editor/Reviewers' comments

Editor's comments:

Dear Prof Li,

Your manuscript entitled "Dual-Transformable, "Trojan-Horse" Nanovehicles Delivering Ultra-Small, Multifunctional, Full Active Pharmaceutical Ingredients Nanotheranostics towards Tumour Eradication" has now been seen by 3 referees, whose comments are appended below. You will see from their comments copied below that while they find your work of considerable potential interest, they have raised quite substantial concerns that must be addressed. In light of these comments, we cannot accept the manuscript for publication, but would be interested in considering a revised version that addresses these serious concerns.

We hope you will find the referees' comments useful as you decide how to proceed. Should further experimental data or analysis allow you to address these criticisms, we would be happy to look at a substantially revised manuscript. However, please bear in mind that we will be reluctant to approach the referees again in the absence of major revisions. If the revision process takes significantly longer than three months, we will be happy to reconsider your paper at a later date, as long as nothing similar has been accepted for publication at Nature Communications or published elsewhere in the meantime.

We are committed to providing a fair and constructive peer-review process. Do not hesitate to contact us if you wish to discuss the revision or if there are specific requests from the reviewers that you believe are technically impossible or unlikely to yield a meaningful outcome.

If you opted into the journal hosting details of a preprint version of your manuscript via a link on our dedicated website (<https://nature-research-under-consideration.nature.com>), it will remain on this site while you are revising your manuscript, as we consider the file to remain active. Should you wish to remove these details, please email naturecommunications@nature.com indicating your manuscript number and the link on our website that was previously sent to you. Please see our pre-publicity policy at <http://www.nature.com/authors/policies/confidentiality.html> For more information, please refer to our FAQ page at <https://nature-research-under-consideration.nature.com/posts/19641-frequently-asked-questions>

When resubmitting your paper, we also ask that you ensure that your manuscript complies with our editorial policies. Specifically, please ensure that the following requirements are met, and any relevant checklists are completed and uploaded as a Related Manuscript file type with the revised article:

Reporting requirements for life sciences research: http://www.nature.com/article-assets/npg/ncomms/authors/ncomms_lifesciences_checklist.pdf

Characterization of chemical and biomolecular materials:

<http://www.nature.com/ncomms/journal-policies/editorial-publishing-policies#Characterization-materials>

Please use the following link to submit your revised manuscript, point-by-point response to the referees' comments (which should be in a separate document to any cover letter) and any completed checklist:

<http://mts-ncomms.nature.com/cgi-bin/main.plex?el=A4S6BBro6A7DmDe1I4A9ftdV8rht113kmecGzIZEM7PggZ>

Please do not hesitate to contact me if you have any questions or would like to discuss the required revisions further. Thank you for the opportunity to review your work.

Best regards,

Dr. Robert Guillatt

Associate Editor

Response: As suggested by the editor, we have performed additional experiments and revised the manuscript to address all the concerns of the reviewers. In order to make the review process convenient, the revision parts were highlighted in red font in the revised manuscript. We responded to the reviewers to specifically address each reviewer's comments point-by-point as follows (“**Figure Rx**” denotes the figures in this Response; “**Figure x**” stands for the data in the manuscript; and “**Figure Sx**” means the results in Supplementary materials).

Reviewers' comments:

Reviewer #1 (Remarks to the Author):

An interesting hierarchically-assembled nanoparticle, pPhD NP, was developed for dual modality tumor imaging and synergistic tumor therapy. The pPhD nanoplatform releases ultra-small therapeutic nanoparticles PhD in weakly acidic tumor microenvironment, then the ultra-small PhD releases pheophorbide a and doxorubicin. The pPhD nanoparticles not only showed tumor accumulation by near-infrared fluorescence imaging and magnetic resonance imaging, but also induced tumor eradication in different models, indicating the theranostic effect of the nanoplatform.

I recommend acceptance after addressing the minor issues:

(1) Please provide more details about how the circulation half-lives of different NPs (pPhD and upPhD) are calculated based on Figure S15. The injected equivalent amount of DOX should be the same for all three treatment arms. Thus, the initial plasma concentration right after injection should be the same as well, correct?

Response: Thanks for the thoughtful and constructive comments. We calculated the standard PK parameters such as maximum concentration (C-max), area under curve (AUC) and circulation half-time (T-half) based on the concentration trajectories of PK study (Figure S21 in the revised supplementary materials) by using Kinetica 5.0. As shown in **Table R1**, two nanoformulations, including pPhD NPs and upPhD NPs, possessed similar T-half that was much longer (7.2 times) than free DOX, indicating that the nanoformulation enabled to largely mitigate the blood clearance of the chemotherapeutic drugs. Between two nanoformulations, the PEGylated nanoparticles (pPhD NPs) exhibited bigger AUC and higher C-max than un-PEGylated nanoparticles, which supported that the PEGylation improved the blood circulation time and minimized the opsonization effect. For the un-PEGylated formulation (upPhD NPs), its C-max was close to that of free DOX but much lower than that of pPhD NPs, indicating that significant amount of un-PEGylated nanoparticles experienced opsonization effect at the first 2 min after i.v. administration, causing rapid elimination of part of nanoparticles. The comparison in C-max values of different formulations demonstrated the protection effect of PEGylation to nanoparticles in blood circulation. After going through the selective elimination, the escaped upPhD NPs exerted the benefits of nanoparticles, exhibiting similar T-half to the pPhD NPs. In contrast, free DOX showed shorter blood circulation time than both nanoformulations, in aspects of C-max, AUC and T-half. In summary, PEGylated nanoformulation (pPhD NPs) was the most efficient to escape from opsonization and exhibit the best PK profiles. The C-max of un-PEGylated PhD NPs was lower compared to PEGylated nanoformulation as less amount of un-PEGylated PhD NPs avoided the opsonization. The higher AUC and T-half values of un-PEGylated PhD NPs compared to the free drug demonstrated the benefits of the nanoformulation after escaping from opsonization. The PK studies demonstrated that the PEGylated nanoparticles (pPhD NPs) exhibited better PK profiles than their un-PEGylated counterpart (upPhD NPs) and the free drug (DOX). We added more detailed discussion in the revised manuscript.

Table R1. The pharmacokinetic parameters of pPhD NPs, upPhD NPs and DOX. The C-max, AUC and T-half were calculated by Kinetica 5.0.

Formulations	C-max ($\mu\text{g/mL}$)	AUC	T-half (min)
pPhD NPs	36.4744	9258	31.8
upPhD NPs	21.3182	5505	32.6
DOX	22.9727	4161	4.4

The initial plasma concentrations of DOX and nanoformulation were supposed to be same. However, it was difficult to draw the blood immediately after the i.v. administration; there inevitably were dead-time between the i.v. injection and the first blood drawing.

(2) "TEM micrograph (Figure 1b) demonstrated that the pPhD NPs were in spherical morphology, within which revealed a cluster of small dark dots. These small dots are believed to be a micellar assembly of PhD monomers, which further self-assembled into larger nanoaggregates through multi-micelle aggregation. "If the larger nanoaggregates are formed by micellular PhD monomers (suggested as small dots), should not we expect a uniform structure? Why does it show a heterogenous dotted structure?"

Response: Thanks for the comments. We proposed that the reasons may possibly be: i). The linker that attached the main structure of Pa and DOX was flexible. In assembled PhD NPs, the linker may not extend at the same level during the self-assembly process, and therefore, the small nanoparticles were not very uniform. Most nanoparticles that self-assembled by single material were not able to form very uniform nanoparticles. The size was likely to be distributed in a reasonable range. Taking the most commonly used DSPE-PEG₂₀₀₀ composed micelles as an example (**Figure R1**), the nanoparticles cannot assemble in very uniform structures either. The reason was also ascribed to the extension degree or the molecular length of the building block. Ref: a). *Nano Lett.*, **13**, 2528-2534 (2013); b) *J Biomed. Nanotech.*, **13**, 534-543 (2017). ii) Some publications reported the same self-assembly process: (1). *J. Am. Chem. Soc.*, **136**, 11748-11756 (2014); (2). *Soft Matter*, **9**, 3293-3304 (2013). The TEM micrographs (**Figure R2**) in these references showed that the small nanoparticles in the large assembly were not in uniform morphology, which further supported that the heterogenous dotted structures commonly occurred in this kind of self-assembly.

Figure R1 The TEM micrographs adapted from a). *Nano Lett.*, **13**, 2528-2534 (2013); b) *J Biomed. Nanotech.*, **13**, 534-543 (2017).

Figure R2 The TEM micrographs adapted from a) *J. Am. Chem. Soc.*, **136**, 11748-11756 (2014); b). *Soft Matter*, **9**, 3293-3304 (2013).

Reviewer #2 (Remarks to the Author):

This manuscript reports study results of the development and evaluation of an interesting nanoparticle drug delivery system for the treatment of oral cancer. This dual size/charge-transformable nanoparticle has a drug delivery ultrasmall nanoparticle (4 nm) encapsulated in a PEG nanoparticle, resulting in a relatively small polymeric double layer nanoparticle carrying two therapeutic agents (pPhD NP). Extensive studies were done to fully characterize this drug delivery carrier. Result and figures are well-presented and support the conclusion. MRI contrast Mn²⁺ chelated pPhD NPs provided MRI capacity and in vivo results showed selective accumulation of pPhD into the tumors. Conditional release of ultra-small drug-nanoparticles from pPhD was also demonstrated. Furthermore, strong anti-tumor effect was found in the tumor bearing mice with the combination of delivery of Dox and photodynamic therapy. The design of the drug carrier is novel and based on special properties in tumor microenvironment to overcome drug delivery barrier.

Response: Thanks for the highly encouraging comments.

Reviewer #3 (Remarks to the Author):

The strategy of using dual size/charge-transformable nanoparticles for drug delivery is well known so as the idea of using drug amphiphiles themselves to form nanoparticles. In this manuscript, authors created a pegylated pheophorbide a-hydrazone-doxorubicin nanoparticles (pPhD NP) by adapting these approaches with a newly synthesized, acid-labile porphyrin-doxorubicin conjugate, of which its porphyrin moiety presents both phototherapy and chemotherapy opportunities and multimodal imaging capabilities, which is also well established. So the novelty is modest. As for the concept, the overall design of pPhD NP for optimizing the ADMET process has merit albeit a very

complex one. Hence the devils are in details in terms of fully supporting all the claims with rigorous experimental design and data. Unfortunately, the manuscript falls well short of this standard.

Some of the key concerns are:

(1) Monomers (conjugates) characterization is not completed. Full characterization including the complete NMR assignments of any new compounds must be presented to meet the minimal ACS standard. There is also no molecular characterization for Phy-Mn²⁺.

Response: We thank the reviewer for pointing out the monomers characterization in the manuscript, as this is very important for newly synthesized compounds. In our experiments, the Pheophorbide a (Pa), doxorubicin (DOX) and dual-hydroxyl terminated PEG2000 were purchased from commercial sources. The commercial companies provided detailed characterization data, including the mass spectrometry (MS), nuclear magnetic resonance (NMR). We used these materials directly without any further purification or modification. The newly synthesized compounds included Pheophorbide a-hydrazone (Phy), Pheophorbide a-hydrazone-DOX (PhD) and dual-aldehydes terminated PEG2000 (PEG-2CHO). According to reviewer's comments, we assigned the NMR peaks for all three new compounds based on ACS standard. The experimental results and detailed descriptions were provided in the revised supplementary data (**Figure S3, S5 and S8**) and experimental section.

For Phy-Mn²⁺, neither MS nor NMR was able to characterize the Mn²⁺ chelation. We looked up more relevant references and found that the characterizations of manganese chelated porphyrin derivatives were mainly based on their optical properties. For instance, **Figure R3** adapted from "*Bioconj. Chem.* **23**, 1726-1730 (2012)" showed the fluorescence quench of pheophorbide after chelation of Mn²⁺; Therefore, the fluorescence quench of pheophorbide after reaction can be considered that the Mn²⁺ was successfully chelated. **Figure R4** adapted from "*Nat. Mater.* **10**, 324–332 (2011)" supported that the UV-vis spectra shift after chelating different metals on pheophorbide. The UV-vis absorbance peak shift can also be considered as the indication of successful chelation of metal ions. In our experiments (**Figure R5**), we compared the UV-vis and fluorescence spectra before and after Mn²⁺ chelation. The UV-vis absorbance and fluorescence of Phy and PhD monomers both changed after being chelated with Mn²⁺. The UV-vis showed obvious spectra shift (**Figure R5a, b**) as that in **Figure R4** and the fluorescence of Phy and PhD were both largely quenched (**Figure R5c, d**) as shown in **Figure R3**. These evidences both supported that the Mn²⁺ was successfully chelated on Phy in the present manuscript. We added more detailed descriptions in the revised manuscript.

Figure R3. Chelation of Mn quenches the fluorescence of pyro-lipid. MnPL and pyro-lipid in methanol at equivalent concentrations. Adapted from “*Bioconj. Chem.* **23**, 1726-1730 (2012)”

Figure R4. UV-vis absorbance of the porphyrin–lipid subunits incorporated in porphyrinsomes formed from pyropheophorbide (blue), zinc-pyropheophorbide (orange) and bacteriochlorophyll (red) in PBS. Adapted from “*Nat. Mater.* **10**, 324–332 (2011)”

Figure R5. (Figure S18 in first edition or Figure S24 in revised edition). Optical measurements of the Phy and PhD monomer before and after chelated with manganese II (Mn^{2+}). a) & b) UV-vis absorbance; c) & d) Fluorescence spectra.

(2) **Nanoparticle characterization is incomplete:** a) despite the claim of ultra small size of 4nm for PhD NP in the abstract, there is no data to support it; b) Please provide the TEM of PhD NPs before and after the coating with PEG and explain the difference between PhD NPs and upPhD NPs (Fig S13 and Fig S14); c) based on the Figure 1a TEM data, size of pPhD NPs is smaller than 50nm, while its DLS showed 79nm with un-symmetric distribution, please explain; d) A cluster of small dark dots in pPhD NPs is not clearly visualized by the TEM image.

Response: a) Thanks for the comments. In the abstract, we claimed that “*In tumour micro-environment, the pPhD NPs responsively transformed to full API nanotheranostics with ultra-small size (~4 nm) and higher surface charge (35 mV), which dramatically facilitated the tumour penetration and cell internalization*”. We actually provided supporting data in **Figure 2a**. **Figure 2a** (0 h) showed the initial state of pPhD NPs, which hived hundreds of small dark dots (PhD NPs). The diameter of these dots was around 4 nm by TEM measurement. Then, the pPhD NPs were treated with slightly acidic pH (6.8) and observed by TEM at different time points. The small dots (PhD NPs) were gradually released out with

time elapse (2 h and 12 h in **Figure 2a**). We also measured the size of the released PhD NPs by TEM, and further confirmed that the PhD NPs showed ultra-small size, with a diameter around 4 nm.

b) Thanks for the insight and constructive comments. Since the reviewer commented “provide the TEM of PhD NPs before and after the coating with PEG”, and pointed out the Figure S13 and Figure S14, we believed that the reviewer meant the differences between the PEGylated PhD NPs (pPhD NPs) and un-PEGylated PhD NPs (upPhD NPs), but not the PhD NPs and upPhD NPs. Firstly, we would like to explain that the self-assembly of upPhD NPs was an instantaneous process, we were not able to capture the intermediate state (the formation of PhD NPs), either by TEM or DLS. The evidence that confirmed the existence of PhD NPs was the TEM micrograph (Figure 2a). The rationale of this kind of two-step self-assembly was similar to that reported in references including: i). *J. Am. Chem. Soc.*, **136**, 11748-11756 (2014); ii). *Soft Matter*, **9**, 3293-3304 (2013); iii). *J. Am. Chem. Soc.*, **139**, 9124-9127 (2017). iv). *Macromolecules*, **38**, 8679–8686 (2005). Based on our experimental results (Figure 2a), the pPhD NPs were constructed by hundreds of small black dots (PhD NPs).

To compare the upPhD NPs and pPhD NPs, we complemented more TEM observations and captured high-resolution images of upPhD NPs (**Figure R6**). In the TEM micrograph, the upPhD NPs showed large nanoparticles constructed with clusters of small-dots (PhD NPs), but without PEG corona wrapping the nanoparticle. The aggregation-density of these small dots in upPhD NPs seemed higher than those in pPhD NPs. We hypothesized that the differences may be caused by the equilibrium of the hydrophobic and hydrophilic constructions of the nanoparticles. In upPhD NPs, the hydrophilic part was doxorubicin that distributed along the rim of the nanostructure; while in pPhD NPs, the hydrophilic part was PEG₂₀₀₀, which was much larger and more hydrophilic than doxorubicin. Therefore, the hydrophilic parts of upPhD NPs were relatively weaker than pPhD NPs and therefore making upPhD NPs pack relatively tighter. The fluorescence behaviours further supported that PhD NPs stacked tighter upPhD NPs. As shown in **Figure R7**, the fluorescence quench of Pa in upPhD NPs was much stronger than that in pPhD NPs (the fluorescence quench was dominated by aggregation caused quench phenomenon). The critical aggregation concentration (CAC) of upPhD NPs also supported our hypothesis. As shown in **Figure R8**, the CAC of un-PEGylated nanoparticles (upPhD NPs) was around 0.5 μM , which is 6 times lower than its PEGylated counterparts (pPhD NPs was 3 μM , Figure S11 in supplementary data). The lower CAC of upPhD NPs suggested that upPhD NPs were much easier to be formed as nano-aggregations. The results from fluorescence quench and CAC studies both co-related to the TEM results, suggesting that small dots (PhD NPs) stacked tighter in upPhD NPs. upPhD NPs and pPhD NPs experienced similar self-assembly process; but the equilibrium of the two nanoformulations was different, since their hydrophilic parts were different. The TEM micrographs and relevant descriptions were added in the revised manuscript.

Figure R6 The TEM micrograph of upPhD NPs. The scale bar is 50 nm.

Figure R7. The fluorescence quench of Pa in upPhD NPs (5 μ M) and pPhD NPs (5 μ M) by comparing with PhD monomer.

Figure R8. The critical aggregation concentrations (CAC) of upPhD NPs.

c) Thanks for the comments. Generally, the size measured by DLS results are larger than that observed by TEM, as the diameters in DLS measurements stand for hydrodynamic diameter (Reference: *Environ. Sci. Technol.* **43**, 7277-7284 (2009)). TEM requires us to observe samples in the dry state (i.e. most compact state), whereas DLS allows us to observe samples in the solvated state where there will be solvent molecules associated with the nanoparticles.

d) Thanks for the comments. To ensure that the small dots can be seen clearly, the image quality of the inset in **Figure 1** was improved. We also provided the high-resolution image of pPhD NPs in **Figure 2a**. In the 0 h images in **Figure 2a**, the small dots can be clearly observed. In the 2 h and 12 h images, the half-release and fully release PhD NPs was also readily to be observed, respectively.

(3) The methods used to measure absorption and fluorescence are not clearly described. It is surprised that the fluorescence (Figure 1d) was measured at such a high concentration (50uM) as fluorescence should be much more sensitive than absorption.

Response: Thanks for the comments. More detailed methods of the optical measurements were added in the method section, including the instrumentation, cuvette specification, excitation/emission bandwidth and data interval, etc. Thanks for reviewer's thoughtful suggestions, we realized that the fluorescence should be evaluated at diluted concentration, as the concentrated samples may suffer from aggregation-caused quench (ACQ) problem. Based on reviewer's comments, the fluorescence spectra of all materials were re-measured, including the spectra of Phy, DOX, PhD monomers and pPhD NPs at 5 μM concentrations (the CAC of pPhD NPs was 3 μM , we were not able to further decrease the concentrations). As shown in **Figure R9**, the fluorescence behaviours were similar to those tested at 50 μM . The related fluorescence spectra were replaced in the revised manuscript.

Figure R9. Fluorescence spectra of Phy, DOX, PhD monomer and pPhD NPs. a) excitation of 488 nm and **b)** excited at 412 nm. The concentrations of all materials were set to 5 μ M.

(4) Missing of a critical control: upPhD NPs should not be called non-transformable nanoformulation as it has no PEGylation coating. A very important non-transformable control is a pH stable PEGylated pPhD NPs (e.g. using PEGylation with stable cross-linked chemistry instead of the formation of Schiff base bond). This control should be used in drug release study (Figure 1F) and the further in vitro and in vivo studies, to prove nanoparticles transformation directed efficient drug delivery.

Response: Thanks for the very insightful comments. Actually, we have previously considered using the non-transformable control that constructed with stable chemical bonds between PEG and PhD NPs as suggested by the reviewer. The stable conjugation could be realized either by using a reductive agent, such as NaCNBH₃, to reduce the Schiff base bond (C=N) to stable chemical bond (C-N) or employed dual-carboxyl group terminated PEG forming amido bond to mount on the surface of upPhD NPs. Since the reductive method may also reduce the hydrazone bond and make Pa and DOX hard to be detached, we used a dual-carboxyl group-terminated PEG₂₀₀₀ with the NHS-ester (O, O'-Bis[2-(N-Succinimidylsuccinylamino) ethyl] polyethylene glycol 2,000, CAS No: 186020-53-1, **Figure R10**) to attach the PEG and PhD NPs through amide bond. The NHS ester could readily react with amine groups on the surface of the nanoparticles without experiencing harsh reaction process.

Figure R10. Chemical structure of O, O'-Bis[2-(N-Succinimidylsuccinylamino) ethyl] polyethylene glycol 2,000 (PEG-2NHS)

The “non-cleavable pPhD NPs” were then fabricated by the same method as that we used to make pPhD NPs. DLS results indicated the “non-cleavable pPhD NPs” were around 87 nm, with a polydispersity of 0.23 (**Figure R11**). The “non-cleavable pPhD NPs” showed similar nanostructures with pPhD NPs (inset of **Figure R11**), that is, a cluster of small nanoparticles hived in a relatively bigger nanostructure. The surface charge was decreased from 42 mV to 9.73 mV (**Figure R12**), indicating that most amine groups on the surface of the nanoparticles were shielded by PEG. The DLS and TEM results supported that the “non-cleavable pPhD NPs” were successfully synthesized.

Figure R11. Size distribution and morphology of “non-cleavable pPhD NPs”. Scale bar is 50 nm.

Figure R12. Surface charge changes of “non-cleavable pPhD NPs” before and after surface modification with “non-cleavable PEG”.

However, after a series of experiments, we realized that the “non-cleavable pPhD NPs” suffered from the following drawbacks to serve as an appropriate control to pPhD NPs:

i) The drug release of “non-cleavable pPhD NPs” was not as efficient as the pPhD NPs (**Figure R13**). The cleavable pPhD NPs enabled to release drug around 80 % after 48 h, but the “non-cleavable pPhD NPs” was only ~40%. The inefficient drug release may be ascribed to the un-detachable PEG cross-linkage. If the efficacy of “non-cleavable pPhD NPs” was lower than pPhD NPs *in vivo*, we cannot differentiate that the lower efficacy was attributed to the non-transformability or the lower drug release efficiency.

Figure R13. Drug release pattern of “non-cleavable pPhD NPs”. Since some DOX attached to a PEG₂₀₀₀ pendant, the drug releases were evaluated by using dialysis tubes with MWCO of 7,000 Da to make sure all released DOX were able to be collected.

ii) The stable chemical bond between PEG and DOX was not able to be cleaved, which made some DOX molecules attaching to a long PEG pendant, leading to the loss of function (LOF) of DOX. Therefore, even though the “non-cleavable pPhD NPs” cannot achieve considerable efficacy *in vivo*, we cannot determine that the lower efficacy was caused by non-transformability or from the LOF of DOX. The LOF of DOX was proved by comparing the cell viability between “non-cleavable pPhD NPs” and pPhD NPs. As shown in **Figure R14**, the “non-cleavable pPhD NPs” exhibited much weaker *in vitro* anti-tumour activities on OSC-3 cells than “cleavable pPhD NPs”, indicating that DOX in “non-cleavable pPhD NPs” was not fully therapeutically active. To exclude the attribute that the inefficient release also contributed to less efficacy of “non-cleavable pPhD NPs”, we mimicked the PEGylation process of the pPhD NPs fabrication and incubated same molar of PEG-2NHS and free DOX in water for 48 h. Since one PEG has two NHS group at each terminal, the molar ratio of NHS to DOX was 2:1. In this case, part of DOX was conjugated to PEG, resulting in DOX modified with PEG-pendant, which was similar to the

surface non-cleavable bond dominated PEGylation. The introduction of PEG-pendant hypothetically led to the LOF of DOX. Therefore, the cytotoxicity of “PEG-2NHS and DOX reaction system” was evaluated in comparison to the same molar of DOX. As shown in **Figure R15**, the *in vitro* anti-tumour efficacy of DOX in “PEG-2NHS and DOX reaction system” was mostly lost. The LOF of DOX in **Figure 14** and **R15** indicated that the “non-cleavable pPhD NPs” was not comparable control to pPhD NPs.

Figure R14. Cytotoxicity of “non-cleavable pPhD NPs”. The OSC-3 cells were treated with pPhD NPs and “Non-cleavable pPhD NPs”. After 12 h treatments, extracellular nanoparticles were washed and replaced with fresh medium. The cells were further incubated for another 24 h. The method of cytotoxicity evaluation was consistent with that described in the manuscript.

Figure R15. The cytotoxicity of PEG-2DOX and DOX on OSC-3 cells. “PEG-2NHS and DOX reaction system” and free DOX were incubated with OSC-3 cells for 24 h, the cell viability was evaluated by MTT assay. The concentrations were calculated based on the DOX.

Based on the evidence above, we believed that the “non-cleavable pPhD NPs” was not an ideal control to pPhD NPs, as neither their drug efficacy nor the drug release was comparable with pPhD NPs. In this work, the superiorities of the size/charge transformability of pPhD NPs were successfully proved in cellular level by mimicking the slightly acidic pH in tumor micro-environment (TME). At *in vitro* level, an ideal control (post-transformed pPhD NPs, by treated pPhD NPs in pH6.8 overnight) was introduced to demonstrate the transformability. The benefits of the dual-transformability were proved by evaluating the cell uptake, ROS production, apoptosis, cell spheroid penetrations, by comparing the post-transformed nanoparticles (pH6.8) and original nanoparticles (pH7.4). These *in vitro* experimental results positively supported the advantages of the dual-transformability. At *in vivo* level, a non-transformable control was needed, as the TME was not able to be adjusted (cannot use post-transformed nanoparticles as control). Therefore, an alternative control was requisite to demonstrate the transformability of the pPhD NPs. After carefully considering the related experimental results, the un-PEGylated PhD NPs (upPhD NPs) were selected as a non-transformable control. The reasons are listed as follows:

(i) The self-assembly of pPhD NPs and upPhD NPs were very similar, both experienced a secondary assembly process. Their morphologies are very close as well (Figure 2a and Figure R6). The only differences between pPhD NPs and upPhD NPs were the PEGylation, which make the transformability of these two types of nanoparticles completely different. Based on the rationale of the self-assembly, the upPhD NPs was not transformable when the surrounding pH changed to TME pH. We hypothetically summarized the possible mechanism of the differences in transformability between pPhD NPs and upPhD NPs. As shown in **Figure R16**, PhD monomer was an amphiphilic molecule and inclined to assemble into micelle-like nanoparticles with the hydrophilic parts facing out and the hydrophobic parts avoiding contacting with water and facing in. Then, the small micelle-like nanoparticles further aggregated into larger nanostructure through multi-micelles aggregation. In upPhD NPs, the hydrophilic parts were the amine groups (or doxorubicin) on the surface of the multi-micelle aggregation. When the pH of the surroundings was changed a little bit (from 7.4 to 6.8), the equilibrium of such big assembly wouldn't change a lot. Therefore, the nanostructure of upPhD NPs didn't collapse in acidic TME. In comparison, the pPhD NPs experienced the same self-assembly process to upPhD NPs. After PEGylation, the hydrophilic-building blocks were changed from DOX to PEG₂₀₀₀ (**Figure R16**). The equilibrium of self-assembly was changed as well, making the density of the PhD NPs aggregation lower (as we discussed in comments 2b). When pPhD NPs encountered the acidic TME, the hydrophilic-building block (PEG₂₀₀₀) detached from the pPhD NPs. At this point the whole nanoparticles suddenly lost the hydrophilic parts, and the equilibrium of the nano-assembly was therefore changed, resulting in the collapse of the pPhD NPs into small PhD NPs.

Figure R16. Schematic illustration of the transformability between Un-PEGylate PhD NPs (upPhD NPs) and PEGylated PhD NPs (pPhD NPs)

(ii) The non-transformability of upPhD NPs was also evaluated by TEM and DLS, as shown in **Figure R17a**. upPhD NPs showed large nanoparticles constructed with clusters of small-dots (PhD NPs), but without PEG outer layer wrapping the nanoparticle compared with pPhD NPs. The density of the small dots was also higher than that in pPhD NPs (see in manuscript Figure 2a), indicating that the interaction between each small dot in pPhD NPs may not as strong as that in upPhD NPs). This may further explain why the pPhD NPs were easier to be transformed into smaller nanoparticles. Moreover, as we responded in “**Comments 2(b)**”, the fluorescence quench of pheophorbide a in pPhD NPs was much weaker than that in upPhD NPs, suggesting that the “pheophorbide a” molecules in pPhD NPs stacked looser than those in upPhD NPs. This corresponded to the TEM results (**Figure R17** and Figure 2a), and further indicated that the interaction between each small dot was stronger in upPhD NPs. According to the DLS and TEM results shown in **Figure R17b**, upPhD NPs stayed in big nano-aggregation state after the surrounding pH was adjusted to a slightly acidic value, which was consistent with our hypothesis and supported that upPhD NPs were a type of non-transformable control.

Figure R17. The TEM micrograph of upPhD NPs in pH7.4 and pH6.8 (treated for 12 h)

(iii) The upPhD NPs showed similar *in vitro* anti-tumour activity to pPhD NPs (**Figure R18**), which makes upPhD NPs match pPhD NPs in the drug efficacy.

Figure R18. The cytotoxicity of upPhD NPs and pPhD NPs towards OSC-3 cells.

iv) The drug release of upPhD NPs was as efficient as that of pPhD NPs (**Figure R19**).

Figure R19. Drug releasing pattern of upPhD NPs.

Based on the discussion above, we believed that upPhD NPs were a better control to pPhD NPs than the stable chemical bond dominated nanoparticles (“non-cleavable pPhD NPs”). **With upPhD NPs as a control, the upPhD NPs and pPhD NPs would exhibit similar *in vivo* efficacy provided they have the same level of tumour accumulation and cell uptake. If the different efficacy presented, the reason can be easily attributed to their final accumulation in cancer cells, but not the LOF of DOX or the inefficient drug release as for “non-cleavable pPhD NPs”.** The benefits of using upPhD NPs as control not only included the size/charge dual-transformability, but also the advantages of the PEGylation. In our animal experiment, we observed the superior efficacy of PEGylated transformable nanoparticles (pPhD NPs) comparing to the non-PEGylated and non-transformable controls, which well supported that the benefits of the dual-transformability. We added more detailed information of upPhD NPs to elaborate this control in the revised manuscript.

(5) Pharmacokinetics measurement should be able to calculate blood circulation half-time to guide the *in vivo* study.

Response: Thanks for the comments. As shown in **Table R1** and the response to reviewer 1, we calculated the C-max, AUC and T-half based on the PK study. More detailed discussion was added in the revised manuscript.

(6) The biodistribution imaging data (Figure S16 and 17) are inconsistent to the *ex vivo* biodistribution data in Figure 3 where the tumor uptake of the nanoparticle was far higher than the other organs.

Response: Thanks for the comments. In Figure S16 and S17, we observed some fluorescence in abdomen area at some timepoints. But at the last timepoint, no obvious fluorescence signal could be detected in the abdomen, indicating that the nanoparticles may be gradually cleared or metabolized from these areas (45 h after administration). But we could still observe strong signal at the tumour region. The *ex vivo* imaging (**Figure 3**) was obtained after the last timepoint of *in vivo* imaging (the mice were sacrificed at 48 h after administration of nanoparticles). So the uptake of the nanoparticle at the tumour sites was far higher than the other organs.

(7) In Figure 3H, the authors used Fluorescence imaging of ROS productions (NIRF ROS probe) to evaluate photodynamic (PDT) efficacy. Did author really believe that a potent PDT efficacy could be induced with light treatment at 0.4 w/cm² for 3 min.

Response: Thanks for the comments. We believed the PDT efficacy could be induced at such light dose level, because we proved the photodynamic effect by assessment of ROS production step by step in this work. In **Figure 1h**, the ROS production at different concentration was evaluated under the light treatment of 0.4 w/cm² for 3 min. Although the concentrations of nanoparticles were as low as 1 μ M, the ROS production was still obviously observed. These experiments supported that the photodynamic effect of pPhD NPs could be successfully triggered at difference photosensitizer concentrations. At cellular level (**Figure 2d**), the ROS production was also attested in the photosensitizer-treated groups in tumour cells, which was much higher than that in control cells. The apoptosis results in **Figure 3e** pertained to photodynamic therapy also proved that the light treatment group showed more apoptosis than those were not treated with light. In animal experiments, the ROS production was clearly indicated by ROS probe via imaging (**Figure 3h**), and the quantitative results (**Figure 3i**) also confirmed that the PDT effect was much higher than PBS treated group. Therefore, our experimental data well supported that the PDT efficacy could be induced at such light dose level.

(8) Figure 3e displayed Mn-pPhD NPs based MRI. In Figure 3J, MRI was used to monitor photo treatment response (tumor shrinkage) for 168h. Was this MRI signal contributed by Mn-pPhD NPs based MRI (last 7days?) or agent free MRI?

Response: Thanks for the comments. In **Figure 3j**, we provided the MRI of tumour before the administration of nanoparticles (0 h). By comparing the images at 120 h and 168 h with that at 0 h, the MRI images after 120 h (including 120 h) should the agent free MRI, as they exhibited similar MRI pattern to that at 0 h. We added more detailed description in the revised manuscript. We apologize for the confusing illustration of “0 h” in **Figure 3j**, which has the same meaning as the timepoint “pre” shown in

Figure 3e. To make it easily understood, we revised this timepoint from “0” to “pre” in the revised manuscript.

(9) Although a synergistic effect of pPhD NPs (photothermal-, photodynamic- and chemotherapies) was claimed in the abstract, Figure 2i data only showed a combination effect.

Response: Thanks for the comments. We provided the evidence of synergistic effect by calculation of the combination index (CI) between phototherapy and chemotherapy. The data was shown in **Figure 2j** (based on **Figure 2i**). If the CI above 1, the antagonistic effect dominates; CI=1 stands for additive effect; CI<1 indicates synergistic effect (References: *Cancer Res* **70**, 440-446 (2010); *Pharmacol. Rev.* **58**, 621-681 (2006)). Based on the results in **Figure 2j**, the CI values at most concentrations were much lower than 1, indicating the strong synergistic effect between phototherapy and chemotherapy in pPhD NPs against oral cancer cells. The antagonistic effect in the lowest concentration (0.1 μ M) and highest concentration (50 μ M) were ascribed to the neglectable efficacy and overwhelming efficacy of both phototherapy and chemotherapy, respectively. There was no significant difference on efficacy between phototherapy and chemotherapy if the concentrations were either too low (both not kill cells) or too high (both killed all cells). We re-checked the combination index results in **Figure 2j** and found that the data was incorrectly calculated (the concentration of the combination group was mistakenly halved). We apologize for our oversight and we provided the new results in the revised manuscript.

Reviewers' Comments:

Reviewer #1:

Remarks to the Author:

The authors have addressed all comments adequately. It can be accepted for publication as it is.

Reviewer #3:

Remarks to the Author:

At its core, the work is built upon several known approaches, therefore its importance, hence the novelty is justified through the dual-transformable approach and the synergistic trimodality therapy (photothermal-, photodynamic- and chemo-therapies). Unfortunately, many of these key points have not been demonstrated despite authors' great effort in responding reviewers' comments and additional data for the revision. Key concerns are listed below:

1) The claim of transformable approach is not supported by the data.

The transformable approach was measured only in solution by incubating pPhD NPs at slightly acidic pH (6.8, pHe condition) over time to monitor particles size change. Without in vitro and in vivo evidence to demonstrate the particles being stable in serum circulation and being transformed to small nanoparticle to enhance intracellular uptake, this claim is not supported. The authors attempted to showcase the advantage of the first transformable step (pHe) by comparing cell uptake of pPhD NPs with post-transformed pPhD NPs (Figure 2c and 2f). This is totally inadequate as there is no transformation kinetics involved. The second step of this transformation (release active API such as phy and Dox) has also not been demonstrated. The drug release analysis method using dialysis cartridge and UV/vis (Figure 1i) is not convincing, and the HPLC-MS must be performed to provide direct evidence on the prodrug activation at acidic intracellular pH (pHi) condition via hydrazone bonds cleavage.

2) Missing or unsuitable controls for in vivo studies.

a) The selected un-PEGylated PhD NPs (upPhD NPs) is not a suitable control for in vivo study to demonstrate enhanced delivery as its surface properties are distinctively different from pPhD NPs (without vs with PEGylation shield; 43mv vs. 12mv of surface charge), which could easily result in completely different PK and biodistribution profiles. As shown in TableS1, both nanoparticles displayed similar short half-life (30min), however, with such short half-life, it is very surprising to see the highest MRI imaging signal at 24 and 48h post injection for both pPhD NPs and upPhD NPs. This does not make sense. Perhaps, the fluorescence method used for PK analysis is problematic. The fluorescence of Phy (Ex=412 nm and Em=680 nm) used for quantifying pPhD NPs and upPhD NPs in blood circulation is quenched in the nanoparticle formulation, and the quenching is nanoparticle-dependent and changeable during the circulation.

b) The single drug controls used (free DOX and Phy) were not in their nanoformulations. This is evident in Table S1 where Dox shows 4.4 min half-life. Therefore, comparing their therapeutics efficacy to evaluate the synergistic effect is incorrect. Doxil (a Dox nanoformulation) would be a much better control.

c) Given the distinct mechanisms PDT and PTT operate, it is important to determine each's role in phototherapy efficacy. The authors claim both PTT and PDT response based on Figure 3 data, this claim is not fully supported given that the laser setting (high power of light 0.4, 0.8 W/cm², short time 3 min treatment) is favorable to PTT than PDT. In fact, the temperature increase was confirmed in Figure 3g. To support the dual response claim, the experimental design requires both PDT and PTT alone controls with corresponding laser settings.

Given these critical flaws, this work is premature for publication.

Response to the comments

Reviewers' comments (NCOMMS-17-26354A):

Reviewer #1 (Remarks to the Author):

The authors have addressed all comments adequately. It can be accepted for publication as it is.

Response: Thanks for the positive comments.

Reviewer #3:

At its core, the work is built upon several known approaches, therefore its importance, hence the novelty is justified through the dual-transformable approach and the synergistic trimodality therapy (photothermal-, photodynamic- and chemo-therapies). Unfortunately, many of these key points have not been demonstrated despite authors' great effort in responding reviewers' comments and additional data for the revision.

Response: Thanks for the constructive and professional comments. We complemented a significant amount of experiments and added ideal controls to demonstrate the key points suggested by the reviewer.

Key concerns are listed below:

1) The claim of transformable approach is not supported by the data.

The transformable approach was measured only in solution by incubating pPhD NPs at slightly acidic pH (6.8, pHe condition) over time to monitor particles size change. Without in vitro and in vivo evidence to demonstrate the particles being stable in serum circulation and being transformed to small nanoparticle to enhance intracellular uptake, this claim is not supported. The authors attempted to showcase the advantage of the first transformable step (pHe) by comparing cell uptake of pPhD NPs with post-transformed pPhD NPs. This is totally inadequate as there is no transformation kinetics involved. The second step of this transformation (release active API such as phy and Dox) has also not been demonstrated. The drug release analysis method using dialysis cartridge and UV/vis (Figure 1i) is not convincing, and the HPLC-MS must be performed to provide direct evidence on the prodrug activation at acidic intracellular pH (pHi) condition via hydrazone bonds cleavage.

Response: Thanks for the constructive comments. To prove the particle stability in serum, we incubated pPhD NPs with serum for two weeks and monitored the size distributions. As shown in **Fig. R1**, pPhD

NPs retained their hydrodynamic size in the presence of serum, and showed no evidence of collapse for two weeks, which supported that pPhD NPs were stable in serum circulation. The *in vitro* cellular uptake and penetration experiments also supported that the pPhD NPs were stable in serum (Fig. 2c & 2f in the manuscript; **Fig. R3 & R4**, will discuss later), as the neutral pH (7.4) did not lead to much deeper penetration and enhanced cellular uptake than their acidic counterparts. In the medium at pH value (6.8), pPhD NPs dynamically transformed, as the penetration and cellular uptake were both increased. These *in vitro* studies further indicated that the pPhD NPs were stable in serum at neutral pH, and could dynamically transform in serum at pH. For *in vivo* demonstration of the dynamic transformability, we utilized high-resolution microscopic imaging at the tissue level to directly observe the penetration and uptake of pPhD NPs in comparison with non-transformable pPhD NPs (NT-pPhD NPs, will be explained in response to the comment2). After being i.v. administrated into tumour-bearing mice, we directly observed the transformability-induced superior penetration and uptake of pPhD NPs in tumour tissue compared with NT-pPhD NPs. As shown in **Fig. R2**, the “green” fluorescence stained the tumour blood vessel, and “red” fluorescence from Phy indicated the distribution of the nanoparticles. The transformable nanoparticles diffused further beyond the tumour blood vessels, they distributed and lit up the tissue throughout the entire tumour. In comparison, their non-transformable counterparts only distributed in the periphery of the blood vessel and lit up the tumour tissue in short distance from the blood vessel. These results indicated that the pPhD NPs enabled to dynamically transform into smaller nanoparticles and penetrate deeply in tumour tissue. We also tried to obtain physical evidence of the *in vivo* transformation. However, it is technically difficult to isolate the nanoparticles from blood or tumour tissues for further test, as the biological system is complicated, and various kinds of biomacromolecules could mess up the tests. The relevant results and discussion were added into the revised manuscript and marked in red font.

Fig. R1. Stability of pPhD NPs via measurements of size changes in the presence of serum over time. The pPhD NPs were incubated with 10% fetal bovine serum in 37 °C. The size distributions were monitored by dynamic light scattering.

Fig. R2. The microscopic distribution of transformable nanoparticles (pPhD NPs) in tumour tissue in comparison with the non-transformable pPhD NPs (NT-pPhD NPs) to demonstrate the superior tumour penetrations enhanced by transformability. The blood vessel (green) was stained by FITC-dextran (70k) while the red fluorescence indicated the distribution of the nanoparticles. The scale bar is 50 μm .

Furthermore, the pPhD NPs were incubated with OSC-3 cells in culture medium at slightly acidic pH (6.8, pHe condition) to investigate the dynamic transformation-induced superior penetration and enhanced cell uptake (This set of experiments were performed together with the post-transformed nanoparticles before, we did not include them because of the crowded space in Fig. 2 in manuscript). By incubating the pPhD NPs with OSC-3 tumour cell spheroids in slightly acidic medium (pH 6.8), such nanoparticles realized superior penetration in tumour cell spheroids (**Fig. R3**) and significantly higher uptake in tumour cells (**Fig. R4**) than their counterparts at neutral pH. These results indicated that the pPhD NPs could dynamically transform into small nanoparticles with highly positive surface charge at pHe condition. The pPhD NPs with dynamic transformability at pHe condition showed slightly weaker penetration and cell uptake than the post-transformed nanoparticles, because the latter ones were already small nanoparticles with highly positive charge and were not necessary to experience such transformation in serum. The relevant discussion and results were added to the revised manuscript and marked in red font.

Fig. R3. The penetrations of pPhD NPs in OSC-3 cell spheroids. The cell spheroids were treated with pPhD NPs (pH 7.4), post-transformed pPhD NPs (the nanoparticles were pre-treated in pH 6.8) and pPhD NPs in pH 6.8 cell culture medium (pH 6.8).

Fig. R4. The enhanced cellular uptake of pPhD NPs in OSC-3 cells. The cells were treated with pPhD NPs (pH7.4), post-transformed pPhD NPs (the nanoparticles were pre-treated in pH6.8) and pPhD NPs in pH6.8 cell culture medium (pH 6.8). *, $p < 0.05$, **, $p < 0.01$.

The drug release *via* hydrazone bonds cleavage was directly proved by liquid chromatography-mass spectrometry (LC-MS). The PhD monomers were incubated at acidic pH (5.0, pH_i condition) overnight and subjected to LC-MS analysis. As shown in **Fig. R5**, the total ion chromatography (TIC) exhibited three main peaks in the upper panel while each peak corresponded to a mass spectrum at the lower panel. The PhD monomers were synthesized by conjugating a hydrophilic doxorubicin (DOX) and hydrophobic pheophorbide a-hydrazide (Phy) through a hydrazone bond. Hydrazone bond is a kind of traceless reversible chemical bond (Ref: *ACS Nano* **2015**, *9*, 2729-2739; *Nat. Commun.* **2015**, *6*, 6650), and the

cleavage of hydrazone bond will yield two original chemicals that were employed to construct the PhD monomers. Hence, three peaks were expected to appear in the TIC spectrum, including the peaks of Phy, DOX, and PhD amphiphile, once the cleavage occurred. The three peaks should be hydrophilic doxorubicin (~544 Da), hydrophobic Phy (~607 Da) and non-cleaved PhD monomers (~1132 Da). In TIC, the doxorubicin was firstly eluted out (#1, 4.70 min), and the corresponding mass spectrum showed peak at 544.1 Da. The second peak was the PhD monomers (#2, 6.56 min), and was correlated to 1132.5 Da in the mass spectrum. The third TIC peak (#3, 8.54 min) was related to 607.2 Da, indicative of the Phy. The LC-MS results supported that the PhD molecules can be cleaved by pHi from hydrazone bond. The LC-MS analysis method and corresponding results were added into the revised manuscript.

Fig. R5. The hydrazone bonds cleavage was investigated by liquid chromatography-mass spectrometry (LC-MS). PhD monomers were treated in pH 5.0 for overnight and subjected to LC-MS analysis.

2) Missing or unsuitable controls for in vivo studies.

a) The selected un-PEGylated PhD NPs (upPhD NPs) is not a suitable control for in vivo study to demonstrate enhanced delivery as its surface properties are distinctively different from pPhD NPs (without vs with PEGylation shield; 43mv vs. 12mv of surface charge), which could easily result in completely different PK and biodistribution profiles. As shown in TableS1, both nanoparticles displayed similar short half-life (30min), however, with such short half-life, it is very surprising to see the highest MRI imaging signal at 24 and 48h post injection for both pPhD NPs and upPhD NPs. This does not make sense. Perhaps, the fluorescence method used for PK analysis is problematic.

The fluorescence of Phy (Ex=412 nm and Em=680 nm) used for quantifying pPhD NPs and upPhD NPs in blood circulation is quenched in the nanoparticle formulation, and the quenching is nanoparticle-dependent and changeable during the circulation.

b) The single drug controls used (free DOX and Phy) were not in their nanoformulations. This is evident in Table S1 where Dox shows 4.4 min half-life. Therefore, comparing their therapeutics efficacy to evaluate the synergistic effect is incorrect. Doxil (a Dox nanofomulation) would be a much better control.

c) Given the distinct mechanisms PDT and PTT operate, it is important to determine each's role in phototherapy efficacy. The authors claim both PTT and PDT response based on Figure 3 data, this claim is not fully supported given that the laser setting (high power of light 0.4, 0.8 W/cm², short time 3 min treatment) is favorable to PTT than PDT. In fact, the temperature increase was confirmed in Figure 3g. To support the dual response claim, the experimental design requires both PDT and PTT alone controls with corresponding laser settings.

Response: Thanks for the constructive comments. As suggested by the reviewer, we conducted a series of new *in vivo* experiments with appropriate controls, such as non-transformable pPhD NPs (NT-pPhD NPs), Doxil and PDT alone. The three points in comments 2 were correlated, and therefore they have been responded together.

The fluorescence of the samples for the standard curve and each experimental group was measured upon the dilution of blood serum solution in access amount of DMSO (Serum:DMSO = 20:80, vol%), so that the fluorescence quenching could be minimized in the PK studies. We apologize that the detailed information was not included in the method section. The details have been added in the revised manuscript. The loss of nanoparticles from blood circulation does not equal to the amounts that accumulate in the tumour sites, because not all the nanoparticles will accumulate in tumours. A large amount of them will first be eliminated by opsonization once they are i.v. injected. The process of opsonization is one of the most important biological barriers to the drug delivery (Ref: *Int. J. Pharm.*, **2006**, 307 (1) 93-102; *Prog. Lipid Res.*, **2003**, 42, 463-478). Once the nanoparticles were i.v. injected, opsonin proteins present in the blood serum quickly bind to the nanoparticles, allowing macrophages of the mononuclear phagocytic system (MPS) to easily recognize and remove these drug delivery systems before they reach the tumour sites. We calculated the initial (α) and second (β) phases of the T-half time for each group, respectively. As listed in **Table R1**, the T-half (α) of pPhD NPs was 31.8 min, this short T-half (α) was attributed to the opsonization-induced elimination of the nanoparticles during the blood circulation. Due to the opsonization, part of injected pPhD NPs may be removed by MPS before they

reach the tumours shortly after the injection. After escaped from the opsonization, there still were considerable amounts of nanoparticles circulating in the blood and lasting for a long time. The T-half (β) of pPhD NPs extended to 1440 min (24 h). Hence, it is understandable that the nanoparticles started to accumulate in tumours at 4 h or even earlier after the i.v. injection. The tumour accumulation gradually escalated and reached the highest level at 24 h. Therefore, the PK should be accurate and reasonable according to the discussion and results above. We added the relevant data and discussion in the revised manuscript.

Table R1. The pharmacokinetic profile of pPhD NPs, upPhD NPs and DOX. The C-max, AUC and T-half were calculated by Kinetica 5.0.

Group	C-max ($\mu\text{g/mL}$)	AUC	T-half (α)	T-half (β)
pPhD NPs	36.4744	9258	31.8 min	1440 min
upPhD NPs	21.3182	5505	32.6 min	1095 min
DOX	22.9727	4161	4.4 min	360 min

According to the review's comments, we developed new non-transformable pPhD NPs (NT-pPhD NPs) by introducing non-cleavable chemical bonds between the PEG and nanoparticles as a more appropriate non-transformable control. The NT-pPhD NPs were fabricated by following the same procedures of pPhD NPs preparation, except that the PEG-2CHO was replaced by PEG-2COOH NHS ester (**Fig. R6**). The PEG-2COOH NHS ester reacted with the amine groups on the surface of upPhD NPs and formed non-cleavable amide bonds. Such PEGylation could not be detached, and thus the resulting nanoparticles were not transformable. The morphology and size distribution of NT-pPhD NPs were investigated and the results were shown in **Fig. R7**. NT-pPhD NPs presented similar morphology as pPhD NPs and exhibited similar hydrodynamic size at around 87 nm. The transformability was also investigated (**Fig. R8**). Neither the size nor the surface charge of NT-pPhD NPs was changed after being stimulated at pHe (pH 6.8). The building blocks, morphology, surface modification, size distribution and surface charge of NT-pPhD NPs were very similar to these of pPhD NPs, indicating that the NT-pPhD NPs were an ideal control to elucidate the importance of dual-transformability *in vivo*.

Fig. R6. Chemical structure of PEG-2COOH NHS ester.

Fig. R7. Size distribution and morphology of NT-pPhD NPs. The inset is TEM micrograph. The scale is 50 nm.

Fig. R8. Size and surface charge changes of NT-pPhD NPs before and after the stimulation of pHe (6.8).

After the ideal control nanoparticles (NT-pPhD NPs) were prepared, we performed a new set of animal studies to demonstrate the advantages of the transformability. According to reviewers' comments, Doxil was introduced as the nanoformulation of the doxorubicin to investigate the chemotherapeutics delivery ability of pPhD NPs. Another control, pPhD NPs with lower light dose, was set as PDT alone control, as the previous laser dose (0.4 w/cm^2 , 3 min) mostly contributed to PTT due to their feverishly elevated temperature (higher than $22 \text{ }^\circ\text{C}$). Followed the same protocol that used in Figure 4 in the manuscript, the OSC-3 cells were implanted to two positions of the flanks or lips of nude mice to establish subcutaneous and orthotopic tumour models, respectively. After the establishment of tumours, the mice were randomly assigned into 5 groups ($n=6$): PBS, Doxil, NT-pPhD NPs, pPhD NPs and pPhD NPs with low light dose. The low laser dose was set to 0.2 w/cm^2 to generate PDT dominant effect as reported previously (Ref.: *ACS Nano* **2013**, 7, 2541-2550). The total laser power was kept at the same level for the low dose (0.2 w/cm^2 , 6 min) and high dose groups (0.4 w/cm^2 , 3 min). The changes in tumour volume of the

subcutaneous model were shown in **Figure R9a**. As the oral cancer is highly malignant, all the tumours in PBS group grew so fast that all mice died within two weeks. The Doxil exhibited excellent anti-tumour activity and could effectively slow down the tumour growth. The efficacy of chemotherapeutic function alone of pPhD NPs without laser irradiation was comparable to that of Doxil (the difference between the two groups was not statistically significant) and such mono-therapy could effectively slow down the tumour progress. The effectiveness of pPhD NPs can be ascribed to their unique dual-transformability. Although they were very similar to pPhD NPs except for the transformability, NT-pPhD NPs only showed marginal efficacy, which was ascribed to their limited penetration in tumours. We further conducted microscopic imaging studies at tissue level to support our claims. The pPhD NPs and NT-pPhD NPs were i.v. injected into mice respectively. 24 h later, the blood vessel was stained by Dextran-FITC, then the tumours were collected and cryo-sliced for tissue level imaging. As shown in **Fig. R2**, NT-pPhD NPs only stayed in the periphery of the tumour blood vessels. In comparison, the pPhD NPs showed superior tumour penetration and lit up the tissue throughout the tumour. The anti-tumour efficacy and tissue penetration both supported that the transformability of nanoparticles could contribute to the therapeutic efficacy. The phototherapy could be introduced to further enhance the efficacy by applying laser irradiation on the tumours. The laser treatment extensively enhanced the anti-tumour efficacy of NT-pPhD NPs, and the tumour shrunk to smaller size, and one mouse in this group was completely cured (**Fig. R9b**). The pPhD NPs showed exceptional efficacy, and all the tumours were ablated and all the animals were finally cured (**Fig. R9b**), which was dramatically more effective than NT-pPhD NPs. The complete cure rate (CCR) for the pPhD NPs group was 100%. The dramatic improvements of laser-induced anti-tumour efficacy were ascribed to the extremely effective photothermal therapy. As shown in **Fig. R10**, the light treatment elevated the temperature for more than $\sim 17^\circ\text{C}$ in NT-pPhD NPs and $\sim 22^\circ\text{C}$ in pPhD NPs treated tumours, which were strong enough to ablate the tumour tissue. Since the PTT was extremely powerful which could completely deluge the photodynamic therapy and chemotherapy, we introduced a low laser dose treatment (0.2 w/cm^2 , 6 min) to pPhD NPs to evaluate the combination of the chemotherapy and PDT. Such low laser dose only led to similar photothermal effect to PBS group with 0.4 w/cm^2 laser (**Fig. R10**), which yielded imperceptible efficacy on the tumours (“PBS+L” in **Fig. R9a**). Due to the presence of photosensitizer, this low laser dose still elicited abundant ROS production for photodynamic therapy (**Fig. R11**). Therefore, such low laser dose treatment would mostly contribute to PDT. As shown in **Fig. R9a**, the PDT alone enabled to extensively improve the efficacy of pPhD NPs by comparing with the non-laser treated pPhD NPs group. pPhD NPs treated with low dose of laser also showed significantly better anti-tumour efficacy than Doxil. In the orthotopic oral cancer model, each group showed similar efficacy as that in subcutaneous model (**Fig. R9c**): Doxil effectively slowed down the tumour progress; NT-pPhD NPs with high laser dose showed excellent efficacy due to the strong PTT

effect; and pPhD NPs with high laser dose exhibited the best anti-tumour efficacy and achieved 100% CCR (**Fig. R9d**). The PDT and chemotherapy of pPhD NPs (0.2 w/cm², 6 min) resulted in excellent anti-tumour efficacy which was much effective than Doxil.

In comparison with the commercially available and well developed Doxil, pPhD NPs showed the comparable chemotherapeutic effect as a mono-therapy approach in OSC-3 oral cancer model. When the PDT was introduced by using low dose of laser, pPhD NPs were more efficacious than Doxil. When photothermal therapy was involved by using high dose of laser, the efficacy of pPhD NPs overwhelmed all other treatments, including Doxil. In comparison with other reported non-PEGylated full API nanoparticles thus far (Ref.: *J. Am. Chem. Soc.* **2014**, *136*, 11748-11756. *RSC Adv.* **2016**, *6*, 12472-12478. *Bioconjug. Chem.* **2015**, *26*, 2497-2506. *Mol. Pharm.* **2016**, *13*, 190-201), pPhD NPs yielded 100% CCR and showed better anti-tumour efficacy so far, due to the excellent multimodal therapeutic functions, PEGylated cross-linkage and dual-transformability. By comparing with the multimodal full API nanoparticles that we developed, such as upPhD NPs in this manuscript and PaIr NPs (*Biomaterials*, **2018**, *161*, 203-215), the anti-tumour efficacy of pPhD NPs was also superior, which can be ascribed to their excellent dual-transformability and PEGylated cross-linkage. pPhD NPs could be conveniently prepared by simply using biocompatible PEG as the only excipient, the DOX and Pa as the APIs. We utilized simple materials, but integrated versatile functionalities in one single smart nanoplatform. With simple introduction of PEG, we realized dual-transformability, which could sophisticatedly balance the contradicts in the drug delivery (see in the introduction part of the manuscript) and largely enhance the anti-tumour efficacy. With phototherapy, pPhD NPs showed enormous potential for the treatment of malignant cancers that are readily accessible to illumination with light, such as oral cancer, bladder cancer, nasal cancer, skin cancer, esophagus cancer, etc.

Fig. R9. *In vivo* efficacy highlighted the benefits of dual-transformability of pPhD NPs. In this set of animal experiment, couples of new control groups were introduced. The nanoformulation of doxorubicin (Doxil), was employed to investigate the chemotherapeutic effect of pPhD NPs. Non-transformable pPhD NPs were introduced to demonstrate the benefits of the dual-transformability. pPhD NPs with low laser dose (0.2 w/cm^2) was added to highlight the photodynamic effect alone. The mice bearing both subcutaneous and orthotopic tumours, were i.v. administrated with PBS, 4.7 mg/kg Doxil (calculated by DOX concentration), 5.3 mg/kg Phy, 10 mg/kg NT-pPhD NPs and 10 mg/kg pPhD NPs (NT-pPhD NPs and pPhD NPs both were calculated based on PhD monomer's concentration), respectively. For the group treated with pPhD NPs at low laser dose, the laser (680 nm) dose was set at 0.2 w/cm^2 for 6 min. The other groups were all set at 0.4 w/cm^2 for 3 min. (a) The tumour volume changes of subcutaneous tumours ($n=6$) after administration of various treatment groups. The black arrows denote the nanoparticles administration, and red ones point out the tumour treated by laser treatments. (b) The complete cure rate (CCR%) of the subcutaneous tumours. (c) The tumour volume changes of orthotopic

tumours (n=6). (d) The CCR (%) of the orthotopic tumours treated with different groups. *, $p < 0.05$; **, $p < 0.01$; ***, $p < 0.001$; *ns*, not significant.

Fig. R10. Photothermal effect of each group on tumour-bearing mice. The mice in PBS, NT-pPhD NPs and pPhD NPs were treated with 0.4 w/cm^2 laser for 3 min, and those in pPhD NPs (0.2) group were treated with 0.2 w/cm^2 laser for 6 min. The PTT experiments were conducted by anesthetizing the mice first, then exposing the tumours under the laser. The body temperature of mice dropped to $\sim 30 \text{ }^\circ\text{C}$ due to the anesthesia, and the elevated temperature (lower than $10 \text{ }^\circ\text{C}$) in PBS and pPhD NPs (0.2), therefore, gave fewer efficacies. *ns*, not significant; **, $p < 0.01$; ***, $p < 0.001$.

Fig. R11. Photodynamic effect of pPhD NPs (0.2) by comparing with PBS. The mice in PBS and pPhD NPs were both treated with 0.2 w/cm^2 laser for 6 min. The ROS production was indicated by NIRF dye, CellROX. The inset was the NIRFI of the tumours. **, $p < 0.01$.

Reviewers' Comments:

Reviewer #3:

Remarks to the Author:

This reviewer is appreciative that the authors took seriously of my previous comments and addressed most key concerns with new data and controls. I believe this is now a fine manuscript that is suitable for this journal.